# SQUEEZE3D: YOUR 3D GENERATION MODEL IS SECRETLY AN EXTREME NEURAL COMPRESSOR

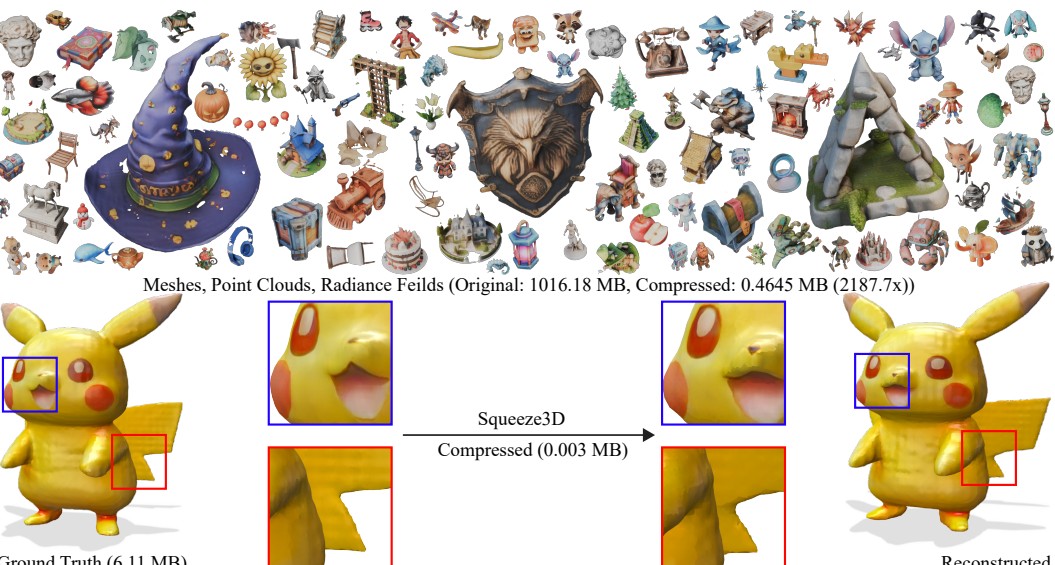

Meshes, Point Clouds, Radiance Feilds (Original: 1016.18 MB, Compressed: 0.4645 MB (2187.7x))

Squeeze3D

Compressed (0.003 MB)

Ground Truth (6.11 MB)

Reconstructed

Figure 1: **Squeeze3D** allows for extreme compression of 3D models while preserving perceptual quality. *Top:* Our method compresses a diverse collection of 3D models. *Bottom:* Comparison between the original model (6.11 MB) and the reconstruction after compression (0.003 MB).

## ABSTRACT

We propose Squeeze3D, a novel framework that leverages implicit prior knowledge learnt by existing pre-trained 3D generative models to compress 3D data at extremely high compression ratios. Our approach bridges the latent spaces between a pre-trained encoder and a pre-trained generation model through trainable mapping networks. Any 3D model represented as a mesh, point cloud, or a radiance field is first encoded by the pre-trained encoder and then transformed (*i.e. compressed*) into a highly compact latent code. This latent code can effectively be used as an extremely compressed representation of the mesh or point cloud. A mapping network transforms the compressed latent code into the latent space of a powerful generative model, which is then conditioned to recreate the original 3D model (*i.e. decompression*). Squeeze3D is trained entirely on generated synthetic data and does not require any 3D datasets. The Squeeze3D architecture can be flexibly used with existing pre-trained 3D encoders and existing generative models. It can flexibly support different formats, including meshes, point clouds, and radiance fields. Our experiments demonstrate that Squeeze3D achieves compression ratios of up to $2187\times$ for textured meshes, $55\times$ for point clouds, and $619\times$ for radiance fields while maintaining visual quality comparable to many existing methods. Squeeze3D only incurs a small compression and decompression latency since it does not involve training object-specific networks to compress an object.

## 1 INTRODUCTION

The rapid advancement of 3D data acquisition and representation technologies over the past decade has significantly expanded the availability and generation of high-resolution 3D content across various domains in different formats, including, meshes, point clouds, and radiance fields (which could be

extracted from a NeRF (Mildenhall et al., 2021) or a 3DGS (Kerbl et al., 2023)). The widespread use of 3D data necessitates the development of techniques that enable efficient transmission, storage, and processing of large-scale 3D representations. To this end, *compression* and the use of *compressed representations* for 3D data are of utmost importance, e.g., in streaming, autonomous navigation, digital twins, remote sensing.

A large body of research proposes techniques to compress meshes, point clouds, neural radiance fields (NeRFs) (Mildenhall et al., 2021), and 3D Gaussian Splats (3DGS) (Kerbl et al., 2023). These approaches aim to maximize the compression ratio while retaining reconstruction quality. For example, traditional mesh decimation techniques (Schroeder et al., 1992; Garland & Heckbert, 1997; Li et al., 2014; Lescoat et al., 2020) remain foundational for polygon reduction, but their reliance on handcrafted simplification rules limits their ability to preserve fine geometric details at extreme compression ratios. MPEG's G-PCC and V-PCC standards (Schwarz et al., 2018; Liu et al., 2019) use projection-based methods for point cloud compression; however, these approaches incur overheads for representing fine details and packing. Prior works also propose a range of compression methods for NeRFs (Li et al., 2024a; Edavamadathil Sivaram et al., 2024; Takikawa et al., 2022; Müller et al., 2022) or 3DGS models (Fan et al., 2024; Lee et al., 2024). Several works propose autoencoder-style networks that compress 3D models into small latent vectors (Zhang et al., 2025; Hahner & Garcke, 2022; Zhou et al., 2020; Chen et al., 2025). Usually, the compression ratios achieved by these methods are of the order of 100x for meshes and the order of 10x for point clouds, but typically much lower.

Our goal is to develop a framework for *extreme* compression of 3D data stored in any format while retaining high visual quality. Recent years have seen significant and continued advances and development of powerful generative models. In this work, we aim to leverage the implicit prior knowledge learnt by the powerful 3D generative models (Xu et al., 2024; Jun & Nichol, 2023) to enable extreme compression ratios. Recent works also propose techniques to leverage generative models for 3D compression (Zhang et al., 2024a; Cui et al., 2024; Zhang et al., 2025; van den Oord et al., 2017; Zhang et al., 2024b; Chang et al., 2025; Roessle et al., 2024) and in one case achieves extreme compression for meshes (Zhang et al., 2024a). However, these approaches require training specialized encoders and generative models for a single 3D format. In contrast, our goal is to flexibly use *existing* encoders and generator models that provide adaptability as encoders/generative models evolve and flexibility across 3D formats.

We propose Squeeze3D, a compression framework, that generates a highly compressed latent vector that can be used to recreate the original 3D data using an existing pre-trained generative model. Squeeze3D comprises three key components: (1) the input 3D data is encoded with a pre-trained encoder. This allows us to extend Squeeze3D to other 3D encoders. (2) We train two small neural networks that we call *forward mapping network* and *reverse mapping network*. The forward mapping network maps the encoded representations into an extremely compressed latent space. The reverse mapping network converts the code from the compressed latent space to the latent space of the generative model. (3) We use a pre-trained generation model to generate the original 3D data using the code generated by the reverse mapping network. Squeeze3D can be flexibly implemented with any pre-trained encoder and generative model.

The forward and reverse mapping networks are trained for any given encoder-generator pair. We first artificially generate a 3D dataset via random prompts to the generator model. This 3D dataset is encoded using the pre-trained encoder. The set of latents produced from the pre-trained encoders (training inputs) and their corresponding latents from the pre-trained generator (ground truth) are used to train the forward and reverse mapping networks. We propose a loss function that minimizes redundant information in the compressed latent space.

Squeeze3D can be flexibly applied to 3D data in different formats. We implement and evaluate our method for mesh, point cloud, and radiance field compression using 3 existing encoders and 5 existing generative models. We demonstrate that our method achieves significantly higher or on-par compression ratios than any existing compression technique for meshes, point clouds, and radiance fields with reconstruction quality that is on par with many prior approaches. We demonstrate a compression ratio of $2187\times$ for a subset of the Objaverse (Deitke et al., 2023) dataset, $55\times$ on a subset of ShapeNet (Chang et al., 2015), $614.9\times$ on a collection of radiance fields (Irshad et al., 2024). While Squeeze3D expectedly cannot achieve state-of-the-art reconstruction quality, we qualitatively show that it is able to retain high visual quality.

**Contributions.** (1) To the best of our knowledge, this is the first framework that leverages *pre-existing pre-trained* generative models to enable extreme compression of 3D data.

(2) We demonstrate the feasibility of establishing correspondences between disparate latent manifolds originating from neural architectures with fundamentally different structures, optimization objectives, and training distributions.

(3) We evaluate Squeeze3D for mesh, point cloud, and radiance field compression and demonstrate that generative models are a promising approach for extreme compression of 3D models. Squeeze3D can be flexibly extended to different encoders, generative models, and 3D formats.

## 2 RELATED WORKS

### 2.1 NEURAL GRAPHIC PRIMITIVES

Neural networks are increasingly being used to represent 2D images (Sitzmann et al., 2020; Tancik et al., 2020; Gao & Jaiman, 2024), 3D objects and scenes (Sitzmann et al., 2019; Jiang et al., 2020; Peng et al., 2020; Tancik et al., 2020; Xie et al., 2022; Davies et al., 2021; Mildenhall et al., 2021; Martel et al., 2021), surface representations (Takikawa et al., 2021; Tang et al., 2022b; Rakotosaona et al., 2023; Edavamadathil Sivaram et al., 2024), occupancy networks (Mescheder et al., 2019; Chen & Zhang, 2019), and signed distance fields (Michalkiewicz et al., 2019; Park et al., 2019; Atzmon & Lipman, 2020). These methods, out of the box, can also be used to compress 3D models in some format since the learned neural network weights are often already significantly smaller. Many NGP compression methods often employ standard neural network techniques to compress MLP by knowledge distillation (Shi & Guillemot, 2022), pruning (Isik, 2021; Xie et al., 2023; Jung et al., 2024), quantization (Shen & McClinton; Zhong et al., 2023; Yang et al., 2023; Gordon et al., 2023; Zhang et al., 2024c), factorizong tensor grids (Gao et al., 2023; Obukhov et al., 2023), low-rank approximation (Shi & Guillemot, 2022; Wang et al., 2024; Tang et al., 2022a; Ji et al., 2025), and using codebooks (Li et al., 2023c;b;a) for quantization. Another approach is to compress feature grids or learnable embeddings (Shin & Park, 2024; Chen et al., 2024a; Pham & Mandt, 2024) or by compressing extracted voxels (Wang et al., 2023; Chen et al., 2022) in contrast to compressing the MLP often. Another set of approaches combines many of these orthogonal improvements to compressing NeRFs (Li et al., 2024b). However, these methods often require training networks per scene or object, incurring significant compression latencies.

### 2.2 3D AUTOENCODERS AND GENERATORS

Autoencoders encode inputs into a latent space and then back into the original input using an encoder-decoder pair; thus, they could be used as compression algorithms. Early approaches relied on volumetric representations, discretizing 3D shapes into voxel grids to leverage ConvNets for encoding and decoding (Liu et al., 2020b; Hess et al., 2023; Li et al., 2023d; Brock et al., 2016; Molnár & Tamás, 2024). While these approaches are effective for regular grid structures, these methods faced scalability challenges due to cubic memory growth with resolution increases. Subsequent advancements focused on spectral methods for encoding 3D shapes in frequency domains, offering compact latent representations but requiring precomputed basis functions that limited generalizability across shape categories (Zamorski et al., 2020; Park et al., 2019).

Several works (Zhao et al., 2023; Chen et al., 2024c; Wu et al., 2024; Lan et al., 2024) use a VAE (Kingma et al., 2013) to compress 3D data into a compact latent which could be used as a compressed representation. There also exist many approaches that train generators to reconstruct radiance fields (Kosiorek et al., 2021; Irshad et al., 2024; Xiang et al., 2024), Gaussians (Wewer et al., 2024; Ma et al., 2024; Xiang et al., 2024), or voxels (Ren et al., 2024; Xiang et al., 2024). Some recent works also pose the problem as learning in a token-space (Chang et al., 2025; Chen et al., 2024b; Siddiqui et al., 2024; Zhang et al., 2024a; 2022; Yang et al., 2024; Luo et al., 2023; Liang & Liang, 2022; Huo et al., 2025; Marques & Da Silva Cruz, 2022; Cui et al., 2023), or with diffusion models (Zhang et al., 2025; Anonymous, 2024; Roessle et al., 2024; Galvis et al., 2024; Ramirez-Jaime et al., 2025; Shao et al., 2024). One such generative model (Zhang et al., 2024a) is able to compress meshes with high compression ratios. Compared to these methods that use autoencoders or generative models for compression, Squeeze3D does not require training specialized encoder-generators for each representation. Instead, Squeeze3D aims to use existing encoders and

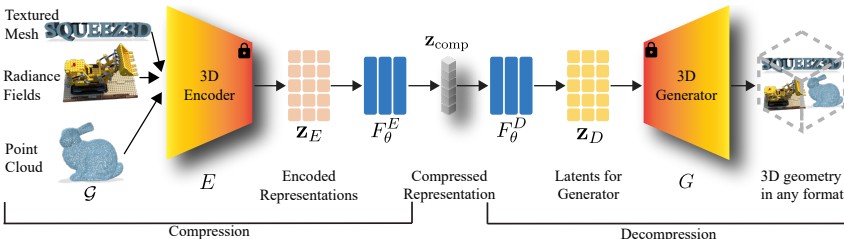

Figure 2: **Overview of our Method.** Squeeze3D bridges arbitrary latent spaces between 3D encoders and generators through trainable mapping networks. During compression, a 3D geometry is encoded and then transformed into a compact representation via the forward mapping network. During decompression, the reverse mapping network converts this representation into the generator's latent space, which is then used to reconstruct the original geometry.

Table 1: **Notation.** The notation we use to describe our method.

| Symb. | Description | Symb. | Description |
|---|---|---|---|
| $\mathcal{G}$ | A 3D geometry in some format | $E$ | 3D encoder model: $E(\mathcal{G}) \mapsto \mathbf{z}_E \in \mathbb{R}^{d_E}$ |
| $G$ | 3D generator model: $G(\mathbf{z}_G, c) \mapsto \mathcal{G}'$ | $\mathbf{z}_E$ | Latent code from the encoder: $\mathbf{z}_E \in \mathbb{R}^{d_E}$ |
| $\mathbf{z}_G$ | Latent code for the generator: $\mathbf{z}_G \in \mathbb{R}^{d_G}$ | $\mathbf{z}_{\text{comp}}$ | Compressed representation $\mathbf{z}_{\text{comp}} \in \mathbb{R}^{d_C}$ |
| $c$ | Conditioning information (*e.g.* text prompt, image) for $G$ | $F_\theta^E$ | Forward mapping network: $F_\theta^E(\mathbf{z}_E) \mapsto \mathbf{z}_{\text{comp}}$ |
| $F_\theta^D$ | Reverse mapping network: $F_\theta^D(\mathbf{z}_{\text{comp}}) \mapsto \mathbf{z}_G$ | $d_C$ | Dimensionality of compressed representation |
| $d_G$ | Dimensionality of generator latent space | $d_E$ | Dimensionality of encoder latent space |

generative models. This enables flexibly adapting the approach as encoders and generative models evolve and supporting different 3D formats.

The work closest to our method is Generative Latent Coding (GLC) (Jia et al., 2024), which trains an autoencoder-style generative model to compress images, particularly compressing the latent representations for an image obtained through VQ-VAE (van den Oord et al., 2017). However, this method is not designed for 3D data.

## 3 METHOD

In this work, we introduce Squeeze3D, a technique to generate highly compressed representations of 3D models by leveraging the implicit prior knowledge learnt by 3D generative models. We also leverage the availability of many 3D encoders to support an extensible set of 3D formats.

The Squeeze3D architecture is depicted in Fig. 2. The key ideas of Squeeze3D are as follows. (1) We leverage existing 3D encoders to generate encoded representations for a given 3D format. Thus, the Squeeze3D architecture can be extended to support different 3D formats by using new or existing encoders. This approach also enables smaller mapping networks, as we now introduce. (2) We use small neural networks to convert from this encoded representation to a highly compressed latent representation (during compression) and then back into the latent space of a 3D generative model (for decompression). We refer to these neural networks as the forward and reverse mapping networks, and they effectively map between the latent space of a 3D encoder to that of a 3D generative model. Thus Squeeze3D can leverage a new 3D generative model by retraining the mapping networks. (3) We propose an additional loss term that enables robust training of the mapping networks to generate a highly compact latent representation that can be used to store/transmit the 3D model. We share an overview of the notation we use to describe our method in Tb. 1.

Mapping networks offers two major benefits over a encoder-generator pair such as MeshAnything (Chen et al., 2024b): (1) The mapping networks typically provide significantly higher compression ratios than existing VAE approaches; (2) Mapping networks provide more flexibility in choice of 3D format, for example, InstantMesh or LRM require multi-view images as input rather than a mesh.

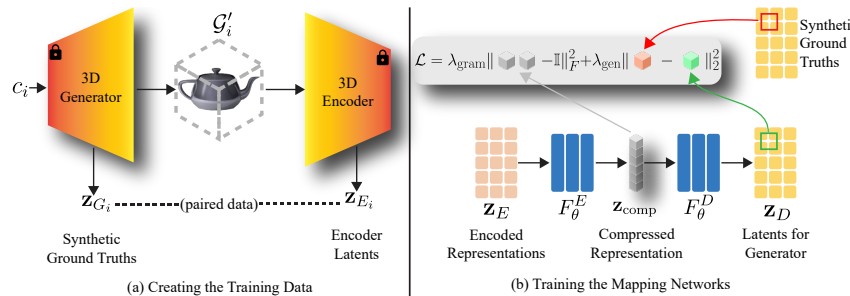

Figure 3: **Training Squeeze3D.** We show an overview of (a) our process of creating synthetic data to train the mapping networks and (b) our process of training the mapping networks.

## 3.1 BRIDGING LATENT SPACES

Squeeze3D comprises two pre-trained models: (1) a 3D encoder $E$ that maps 3D representations to a latent space, and (2) a 3D generator $G$ that synthesizes 3D models of the same initial representation. For a given 3D representation (mesh, point cloud, radiance field) $\mathcal{G}$, the pre-trained encoder $E$ produces a latent representation

$$\mathbf{z}_E = E(\mathcal{G}) \in \mathbb{R}^{d_E} \tag{1}$$

This latent $\mathbf{z}_E$ encapsulates $\mathcal{G}$, but is not in a highly-compressed format. Additionally, we cannot directly use this representation with the generator $G$, as $G$ operates in a different latent space. The generator $G$ synthesizes a 3D model $\mathcal{G}'$ given a latent code $\mathbf{z}_G$ and in some cases conditioning information $c$, $\mathcal{G}' = G(\mathbf{z}_G, c)$. To bridge these disparate latent spaces, we train two mapping networks:

**Forward Mapping network** $F_\theta^E$: Maps from the encoder's latent space to the compressed space, $\mathbf{z}_{\mathrm{comp}} = F_\theta^E(\mathbf{z}_E)$. **Reverse Mapping networks** $F_\theta^D$: Maps from the compressed space to the generator's latent space, $\mathbf{z}_G = F_\theta^D(\mathbf{z}_{\mathrm{comp}})$.

We train $F_\theta^E$, and $F_\theta^D$ together and keep the encoder $E$ and generator $G$ networks frozen.

**Compression.** To compress any 3D model $\mathcal{G}$, the model is first encoded using the pre-trained encoder $E$ and then mapped into a highly compressed latent $\mathbf{z}_{\mathrm{comp}}$ using the forward mapping network $F_\theta^E$

$$\mathbf{z}_{\mathrm{comp}} = F_\theta^E\left(E\left(\mathcal{G}\right)\right) \tag{2}$$

**Decompression.** To decompress the model from its highly compressed latent representation $\mathbf{z}_{\mathrm{comp}}$, we use the reverse mapping network $F_\theta^D$ to obtain the latent in the 3D generator space. The 3D generator then reconstructs the original 3D model $\mathcal{G}(\mathcal{G}')$

$$\mathcal{G}' = G\left(F_\theta^D\left(\mathbf{z}_G\right)\right) \tag{3}$$

## 3.2 TRAINING SQUEEZE3D

In order to train the Squeeze3D architecture, we need to train the mapping networks for any given pair of pre-trained 3D encoders and pre-trained 3D generator models. To train these mapping networks, we need training samples from the encoder's latent space and the corresponding latents in the generator's latent space. These latents in the generator's latent space serve as "ground truth" samples during the training process. We summarize our training process in Fig. 3. We now describe how to generate a training dataset with samples from both of these latent spaces.

Given a pre-trained 3D generator $G$ and encoder $E$, we first sample a diverse collection of conditioning inputs $\mathcal{C} = \{c_i\}_{i=1}^N$ appropriate for the generator model (*e.g.* text prompts for text-to-3D generators, images for image-to-3D generators, or random noise for unconditional generators). For each conditioning input $c_i$, we sample latent vectors $\mathbf{z}_{G_i}$ from the generator $G$ and then synthesize a 3D model using the generator: $\mathcal{G}'i = G(\mathbf{z}_{G_i}, c_i)$. Then we encode this synthetic or generated 3D model using the pre-trained encoder $E$, $\mathbf{z}_{E_i} = E(\mathcal{G}'_i)$. This methodology gives us paired synthetic

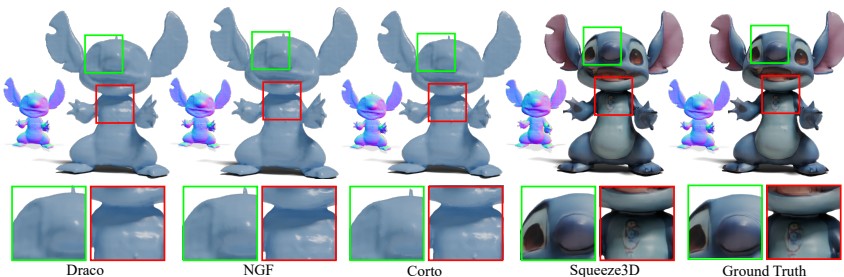

Figure 4: **Qualitative mesh compression results.** We compare Squeeze3D to state-of-the-art methods. Our approach maintains visually important geometric details. Additional results in § C.

data, i.e., latents, for any given pair of encoder and generator. $\{(\mathbf{z}_{E_i}, \mathbf{z}_{G_i})\}_{i=1}^N$, which provides the necessary supervision for training our mapping networks.

The mapping networks $F_\theta^E$ and $F_\theta^D$ together with the generated dataset using the loss shown in **??**.

$$\mathcal{L} = \underbrace{\lambda_{\text{gram}} \| F_\theta^E(\mathbf{z}_E) F_\theta^E(\mathbf{z}_E)^\top - \mathbb{I} \|_F^2}_{\text{orthogonality of compressed representation}} + \underbrace{\lambda_{\text{gen}} \| F_\theta^D \left( F_\theta^E(\mathbf{z}_E) \right) - \text{GT} \|_2^2}_{\text{reconstruction loss}},$$

where GT represents the synthetic ground-truth latents and $\| \cdot \|_F$ denotes the Frobenius norm.

The loss function includes an *reconstruction loss* term that allows us to minimize the difference between the generated latents and the corresponding ground-truth latents. We also add another term, which we refer to as *gram loss*. When training Squeeze3D with only the reconstruction loss term, we found that they concentrate information along a small subset of dimensions, effectively rendering many dimensions redundant.

To understand this, we empirically analyzed the latent vectors $\mathbf{z}_{\text{comp}} = F_\theta^E(\mathbf{z}_E) \in \mathbb{R}^{d_C}$ produced by our forward mapping network. For any batch of size $B$, of encoded 3D models $\{\mathbf{z}_{E_i}\}_{i=1}^B$, we can compute the matrix $\mathbf{Z} \in \mathbb{R}^{B \times d_C}$ where each row is $F_\theta^E(\mathbf{z}_{E_i})$. First, we observe that if we do a singular value decomposition $\mathbf{Z} = \mathbf{U}\boldsymbol{\Sigma}\mathbf{V}^\top$, the singular values in $\Sigma = \text{diag}(\sigma_1, \sigma_2, \ldots, \sigma_{d_C})$ exhibits an extremely skewed distribution: $\sigma_1 \gg \sigma_2 \gg \ldots \gg \sigma_{d_C}$ where $\sigma_i$ represents the $i$-th singular value of $\mathbf{Z}$, arranged in descending order. The condition number $\kappa = \frac{\sigma_1}{\sigma_{d_C}}$ is typically very large, indicating that the effective rank of $\mathbf{Z}$ is much lower than $d_C$.

Second, we observe that the correlation matrix $\mathbf{C} = \frac{1}{B}\mathbf{Z}^\top\mathbf{Z} \in \mathbb{R}^{d_C \times d_C}$ has many off-diagonal elements with large magnitudes in comparison with diagonal elements. This indicates that the dimensions of the compressed representation encode redundant information. Particularly,

$$d_{\text{eff}} = \frac{(\sum_{i=1}^{d_C} \lambda_i)^2}{\sum_{i=1}^{d_C} \lambda_i^2} \ll d_C, \tag{4}$$

where $\lambda_i$ are the eigenvalues. These observations indicate that most of the information in the compressed representation was concentrated along a few dominant directions, with most dimensions contributing negligibly. For compression, this represents a severe inefficiency in utilizing the available parameter budget.

To address this, we propose the Gram loss term that is computed on the outputs of the first mapping network $F_\theta^E$ to force the outputs of $F_\theta^E$ to be orthonormal. A semi-orthogonal matrix $\mathbf{A} \in \mathbb{R}^{m \times n}$ is defined as a matrix that satisfies either $\mathbf{A}\mathbf{A}^\top = \mathbf{I}_m$ (if $m \leq n$) or $\mathbf{A}^\top\mathbf{A} = \mathbf{I}_n$ (if $n \leq m$). Our gram loss term when minimized forces $F_\theta^E(\mathbf{z}_E)F_\theta^E(\mathbf{z}_E)^\top \approx \mathbb{I}$, which is precisely the condition for $F_\theta^E(\mathbf{z}_E)$ to be a semi-orthogonal matrix (when $d_C \leq d_E$).

## 4 EXPERIMENTS

### 4.1 EXPERIMENTAL SETUP

We train Squeeze3D to compress three 3D formats: textured 3D meshes, point clouds, and radiance fields *i.e.* grids of (rgb$\sigma$). For *3D meshes*, we train our approach with MeshAnything (Chen et al.,

2024b) as the encoder and train mapping networks for three 3D generators: Shap-E (Jun & Nichol, 2023), OpenLRM (Hong et al., 2024; He & Wang, 2023), and InstantMesh (Xu et al., 2024). For *point clouds*, we train mapping networks for PointNet++ (Qi et al., 2017) as the encoder and LION (Zeng et al., 2022) as the decoder. For *radiance fields*, we train mapping networks for NeRF-MAE (Irshad et al., 2024) as the encoder and the generator. We compress radiance fields for evaluation to use existing generation models such as (Irshad et al., 2024) that only generate radiance fields in this format. We present additional implementation details in §4.1.

**Training Dataset Creation.** We use the method described in §3.2 to train Squeeze3D for each of our evaluated encoder-generator pair. We now list the datasets that were used to create these latent training datasets.

**Shap-E (Jun & Nichol, 2023) as the generator.** We build a list of 2500 prompts using LLaMA3, each of which is used four times to build a dataset of 10,000 objects (details in the supplementary).

**LRM (Hong et al., 2024; He & Wang, 2023) or InstantMesh (Xu et al., 2024) as the generator.** We rendered 10,000 random objects from Objaverse (Deitke et al., 2023) which serve as image conditions. We also make sure that our rendering follows any conventions the 3D generator expects the input images to follow, for instance, white backgrounds or no backgrounds.

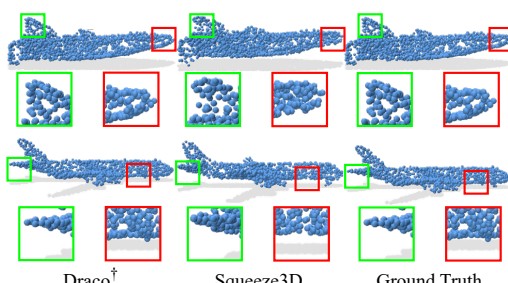

Draco$^\dagger$      Squeeze3D      Ground Truth

Figure 5: **Qualitative point cloud compression results.** We show qualitative results comparing Squeeze3D to state-of-the-art methods. Our approach achieves significantly higher compression ratios while maintaining perceptually important geometric details.

**LION (Zeng et al., 2022) as the generator.** We were able to generate plausible point clouds of 3D objects without any conditioning data from random noise.

**NeRF-MAE (Irshad et al., 2024) as the generator.** We generate radiance fields from the NeRF-MAE (Irshad et al., 2024) dataset.

While many other compression methods require the 3D objects to have certain properties like watertightness, we do not impose any such constraints. We split the datasets into training (80%), and validation (10%) sets generated from our approach. We also built a test (10%) set ($\mathcal{T}$-set) of objects from the datasets: Objaverse, ShapeNet, and NeRF-MAE, on which we report our metrics. Objaverse does not have pre-specified train and test splits. While, our train and test split are specified based on the splits used by MeshAnything Chen et al. (2024b), we do not have access to the splits used for training the generation models. This could cause the test set we build to be a part of the training set of the generation models we use. Thus, we separately construct another test set ($\mathcal{O}$-set) of 500 3D meshes (see §B.2 for the collection process) which are not in the Objaverse dataset, and report results on this test set.

**Evaluation metrics.** We report the standard widely-used metrics, PSNR ↑, MS-SSIM ↑, and LPIPS ↓ (Zhang et al., 2018) for reconstruction quality of meshes and radiance fields. We report standard metrics, PCQM ↑ (Meynet et al., 2020), and PointSSIM ↑ (Alexiou & Ebrahimi, 2020) for reconstruction quality of point clouds. For all the baselines we compare against, we report average compression ratios, as well as compression and decompression times.

### 4.2 3D MESH COMPRESSION

We compare Squeeze3D applied to mesh compression with existing approaches in Tb. 2 (see §A for additional metrics) on our test sets (§4.1). While (Zhang et al., 2023) achieves very high compression ratios on 3D meshes, we are unable to compare with it due to the absence of code and models, and thus qualitatively contrast in §2.2. We make the following observations.

First, Squeeze3D achieves a *mean* compression ratio of **2187**× (6.43 MB → 3 kB), compared to state-of-the-art compression methods: DeepSDF (Park et al., 2019), by more than an order of magnitude (131×, 6.43 MB → 49 kB). Despite this extreme compactness, Squeeze3D preserves

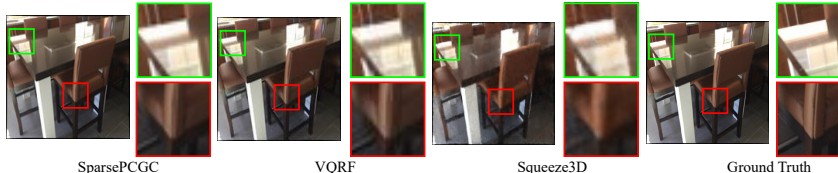

SparsePCGC       VQRF       Squeeze3D       Ground Truth

Figure 6: **Qualitative radiance field compression results.** We show qualitative results comparing Squeeze3D to state-of-the-art methods. Our approach achieves a significantly higher compression ratio while maintaining visually important geometric details.

Table 2: **Mesh Compression.** Quantitative comparison of Squeeze3D with state-of-the-art 3D mesh compression methods. We report compression ratio (CR), compression and decompression times, and quality metrics (PSNR, MS-SSIM, and LPIPS). $\mathcal{T}$ denotes our Objaverse test set, and $\mathcal{O}$ denotes the separately collected set.

| Method | CR (×) (MB) ↑ | $\mathcal{T}$-PSNR ↑ | $\mathcal{T}$-MS-SSIM ↑ | $\mathcal{T}$-LPIPS ↓ | $\mathcal{O}$-PSNR ↑ | $\mathcal{O}$-MS-SSIM ↑ | $\mathcal{O}$-LPIPS ↓ |
|---|---|---|---|---|---|---|---|
| Draco* (Galligan et al., 2018) + JPEG | 6.92 (6.43 / 0.93) | 15.45 (±1.27) | 0.7468 (±0.09) | 0.2437 (±0.09) | 15.40 (±1.30) | 0.7450 (±0.10) | 0.2450 (±0.10) |
| Draco† (Galligan et al., 2018) + JPEG | 6.70 (6.43 / 0.96) | 23.33 (±1.32) | 0.9576 (±0.03) | 0.1039 (±0.06) | 23.30 (±1.35) | 0.9570 (±0.03) | 0.1045 (±0.06) |
| Draco‡ (Galligan et al., 2018) + JPEG | 6.61 (6.43 / 0.97) | 38.91 (±1.30) | 0.9992 (±0.00) | 0.0045 (±0.00) | 38.89 (±1.30) | 0.9991 (±0.00) | 0.0046 (±0.00) |
| Draco§ (Galligan et al., 2018) + JPEG | 6.20 (6.43 / 1.04) | 48.55 (±1.54) | 1.0000 (±0.00) | 0.0004 (±0.00) | 48.50 (±1.55) | 1.0000 (±0.00) | 0.0005 (±0.00) |
| Corto (Lab, 2025) + JPEG | 45.93 (6.43 / 0.14) | 20.92 (±2.79) | 0.8619 (±0.08) | 0.1374 (±0.08) | 20.85 (±2.80) | 0.8610 (±0.08) | 0.1380 (±0.08) |
| Neural Subd. (Liu et al., 2020a) + JPEG | 11.28 (6.43 / 0.57) | 15.95 (±2.18) | 0.8525 (±0.04) | 0.1513 (±0.05) | 15.90 (±2.20) | 0.8520 (±0.04) | 0.1520 (±0.05) |
| DeepSDF (Park et al., 2019) + JPEG | 131.22 (6.43 / 0.05) | 8.47 (±0.23) | 0.7039 (±0.07) | 0.3704 (±0.08) | 8.21 (±0.25) | 0.6850 (±0.08) | 0.3912 (±0.09) |
| NGF (Edavamadathil Sivaram et al., 2024) + JPEG | 42.87 (6.43 / 0.15) | 35.45 (±3.02) | 0.9987 (±0.08) | 0.0054 (±0.03) | 35.40 (±3.05) | 0.9985 (±0.08) | 0.0056 (±0.03) |
| Squeeze3D (InstantMesh) | 2187.07 (6.43 / 0.003) | 27.50 (±3.13) | 0.9796 (±0.02) | 0.0274 (±0.02) | 26.95 (±3.20) | 0.9905 (±0.01) | 0.0124 (±0.01) |

perceptual quality, achieving an LPIPS of 0.0274 versus 0.3704 for DeepSDF which completely fails to reconstruct complex large meshes.

Second, Squeeze3D achieves similar reconstruction quality (LPIPS of 0.0274) as that of approaches such as Draco* (LPIPS of 0.1039), Draco† (LPIPS of 0.0397), and Corto (Lab, 2025) (LPIPS of 0.1374). Squeeze3D also achieves better quality than Neural Subdivision (Liu et al., 2020a) and DeepSDF (Park et al., 2019). Though compared to the state-of-the-art non-learned mesh compression, Squeeze3D cannot achieve as high reconstruction quality, we note that Squeeze3D achieves a significantly higher compression ratio than these approaches. Neural Geometry Fields (Edavamadathil Sivaram et al., 2024) performs better in terms of quality and compression ratios than non-learned methods but does significantly worse in terms of compression size when compared with our approach. We conclude that in achieving very high compression rates, Squeeze3D offers the highest reconstruction quality. Thus, we demonstrate that leveraging 3D generative models is a promising approach for 3D compression.

While using Squeeze3D may not be as fast as some non-learned approaches like Draco (Galligan et al., 2018) or Corto (Lab, 2025), Squeeze3D is often faster than other learned methods. Squeeze3D is particularly much faster than training a network per object, like in NGF (Edavamadathil Sivaram et al., 2024). NGF takes on average 152638 ms to compress objects and 507 ms to decompress objects, compared to 270 ms to compress an object and 1476 ms to decompress objects for Squeeze3D. We qualitatively compare against Draco, Corto, and NGF for compressing meshes in Fig. 4. We note that the results from our approach retain high visual quality due to the use of priors from a generative model.

## 4.3 3D POINT CLOUD COMPRESSION

We compare our method applied to 3D point cloud compression with previous approaches in Tb. 3 (see §A for additional metrics). We notice that our approach achieves a significantly higher compression ratio of 117 (117 / 1.00) opposed to 22.41 (117 / 5.22) by previous approaches. Our approach, while achieving significantly higher compression ratios, leads to only a 0.6898 lower PCQM. We show qualitative results in Fig. 5.

## 4.4 RADIANCE FIELD COMPRESSION

We compare Squeeze3D applied to radiance field compression with SparsePCGC (Wang et al., 2023) and VQRF (Li et al., 2023a) in Tb. 4. We choose these approaches to compare against since these works (or a setting of these works), akin to our setup, only require (rgb$\sigma$) grids. We notice that our

Table 3: **Point Cloud Compression.** Quantitative comparision of Squeeze3D with state-of-the-art point cloud compression methods. We report compression ratio (CR), compression and decompression times, and quality metrics (PCQM and PointSSIM).

| Method | CR (×) (KB) ↑ | Compress (ms) ↓ | Decompress (ms) ↓ | PCQM ↑ | PointSSIM ↑ |
|---|---|---|---|---|---|
| Draco[‡] (Galligan et al., 2018) | 15.64 (117 / 7.48) | 0.35 | 0.34 | 3.2875 (± 0.13) | 0.9722 (± 0.11) |
| Draco[§] (Galligan et al., 2018) | 22.41 (117 / 5.22) | 0.25 | 0.23 | 2.1039 (± 0.08) | 0.9535 (± 0.12) |
| G-PCC (Schwarz et al., 2018) | 37.21 (117 / 3.15) | 19.76 | 29.92 | 2.8163 (± 0.10) | 0.9331 (± 0.09) |
| V-PCC Liu et al. (2019) | 50.17 (117 / 2.33) | 2.33 | 234.85 | 48.22 (±0.13) | 0.9437 (±0.09) |
| Squeeze3D (LION) | 58.50 (117 / 2.00) | 3.85 | 12.74 | 1.8437 (±1.13) | 0.4484 (±0.11) |

Table 4: **Radiance Field Compression.** Quantitative comparison of Squeeze3D with state-of-the-art 3D radiance field compression methods. We report compression ratio (CR), compression and decompression times, and quality metrics (PSNR, MS-SSIM, and LPIPS).

| Method | CR (×) (MB) ↑ | Compress (ms) ↓ | Decompress (ms) ↓ | PSNR ↑ | MS-SSIM ↑ | LPIPS ↓ |
|---|---|---|---|---|---|---|
| SparsePCGC (Wang et al., 2023) | 78.92 (58.07 / 0.74) | 301.39 | 680.82 | 22.26 ± 0.92 | 0.8947 ± 0.02 | 0.1400 ± 0.02 |
| VQRF (Li et al., 2023a) | 40.25 (58.07 / 1.45) | 120.14 | 20.58 | 29.55 ± 0.01 | 0.9749 ± 0.00 | 0.0618 ± 0.00 |
| Squeeze3D (NeRF-MAE) | 619.41 (58.07 / 0.09) | 45.63 | 75.80 | 26.62 ± 2.57 | 0.9533 ± 0.04 | 0.0743 ± 0.02 |

approach achieves a significantly higher compression ratio of $619\times$ opposed to $40\times$ by previous approaches. Squeeze3D achieves these significantly higher compression rates with only a 0.0125 drop in LPIPS. We show qualitative results in Fig. 6.

## 5 DISCUSSION AND LIMITATIONS

The most significant limitation of our approach is its inherent dependency on the quality and expressiveness of the underlying 3D generative model. The decompressed outputs from Squeeze3D fundamentally cannot exceed the quality of what the generator can produce, as the generator serves as both a prior knowledge base and a quality ceiling. In our evaluation, we observed that reconstruction fidelity is directly correlated with the generative capabilities of the chosen generator model. For instance, when using the Shap-E (Jun & Nichol, 2023) generator, we found it challenging to faithfully reproduce highly complex objects due to limitations in the model's semantic capacity (Appendix C). This limitation, however, positions Squeeze3D to benefit automatically from future advancements in 3D generative modeling.

Since Squeeze3D is designed as a compression-decompression framework that should not require per-scene retraining, the generalization capability of our mapping networks to unseen 3D models is crucial. We evaluated this aspect by testing on a dataset of 158 3D meshes and 227 radiance fields, which served as out-of-distribution samples, and found that our approach maintains consistent performance in Appendix A. Nevertheless, for certain outlier cases where the input 3D model contains features far from the distribution seen during training, compression quality may degrade. Thus, for Squeeze3D to be effectively deployed in practical applications, a fallback mechanism would be beneficial to handle such outlier cases. This could involve either a hybrid approach that combines our method with traditional compression techniques or an adaptive system that detects when the mapping networks are likely to produce low-quality results and switches to alternative compression methods.

## 6 CONCLUSION

In this work, we introduce Squeeze3D, a novel framework for 3D compression that leverages the rich priors contained within existing 3D generation models. Our approach bridges arbitrary latent spaces between different models, enabling unprecedented compression ratios while maintaining high visual fidelity. The key benefit of our approach lies in its ability to use the semantic information already encoded in pre-trained 3D generation models, effectively serving as a neural compressor. By training networks that map between latent spaces, we eliminate the need for per-scene optimization that plagues many neural compression methods. This results in faster compression and decompression times compared to many other learned approaches while maintaining competitive visual quality metrics. We hope this work inspires more research on using generative models for compressing 3D data.

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

APPENDIX CONTENTS

## A  ADDITIONAL IMPLEMENTATION DETAILS

We share additional implementation details to reproduce Squeeze3D.

### A.1  NETWORK DESIGN

For each of the encoder-decoder pairs, we train the mapping networks, which are feed-forward neural networks. We summarize network architectures in Fig. 7. Our architectures for different representations are highly similar.

**Meshes.**  The mapping networks together first flatten the input and apply a linear layer to project it into a hidden dimension. This is followed by LayerNorm (Ba et al., 2016), a GELU nonlinearity (Hendrycks & Gimpel, 2023), and dropout (Srivastava et al., 2014). A second linear layer then produces another hidden representation, which is added residually to the output of

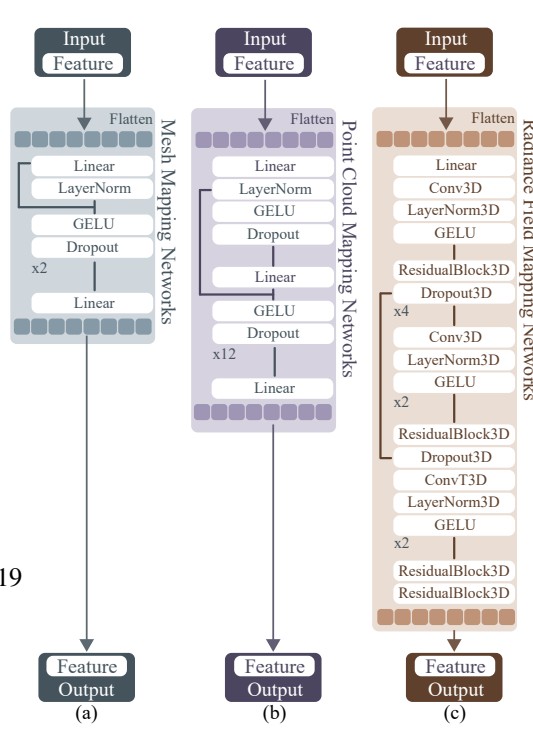

Figure 7: **Network Architectures.**

Table 5: **Model Sizes.** We show the combined size of the forward mapping network and the reverse mapping network we train for each setting. We denote the compressed size in parentheses $(\cdot)$ for the point cloud methods.

| Encoder-Decoder Pair | Parameters (M) |
|---|---|
| MeshAnything (Chen et al., 2024b)-InstantMesh (Xu et al., 2024) | 96.12 |
| MeshAnything (Chen et al., 2024b)-Open LRM (Hong et al., 2024; He & Wang, 2023) | 87.51 |
| MeshAnything (Chen et al., 2024b)-Shap-E (Jun & Nichol, 2023) | 134.53 |
| PointNet++ (Qi et al., 2017)-LION (Zeng et al., 2022) (1024) | 2.11 |
| PointNet++ (Qi et al., 2017)-LION (Zeng et al., 2022) (2048) | 6.53 |
| PointNet++ (Qi et al., 2017)-LION (Zeng et al., 2022) (4096) | 22.29 |
| PointNet++ (Qi et al., 2017)-LION (Zeng et al., 2022) (8192) | 81.48 |
| NeRF-MAE (Irshad et al., 2024) | 86.46 |

Table 6: **Latent Space Sizes.** Sizes of latent spaces for encoder and generator models we use.

| Encoder/Generator Model | Latent Space Size |
|---|---|
| MeshAnything (Chen et al., 2024b) | $257 \times 1024$ |
| InstantMesh TriPlane Features (Xu et al., 2024) | 263168 |
| LION (Zeng et al., 2022) | 8320 |
| PointNet++ (Qi et al., 2017) | 1024 |
| NeRF-MAE (Irshad et al., 2024) | $256 \times 20 \times 20 \times 20$ |

the first linear layer and passed through a second LayerNorm. After another GELU and dropout, a final linear transformation projects into the latent space.

**Point Clouds.** The mapping networks together consist of flattening the input, a linear projection feeds into a LayerNorm (Ba et al., 2016); this normalised vector is stored as a global residual. The sequence then passes through GELU-activated (Hendrycks & Gimpel, 2023) hidden layers of uniform width, each followed by dropout regularisation on the activations (Srivastava et al., 2014). We use two skip-connection schemes: (i) the global residual is re-added immediately after the first hidden layer, and (ii) every fourth hidden layer receives a local residual that adds its own input to its output, creating short four-layer paths. A final linear projection transforms the resulting representation into the target latent space.

**Radiance Fields.** The mapping networks together consist of a custom LayerNorm that first permutes tensors so that LayerNorm (Ba et al., 2016) can operate across channels before restoring the usual NCDHW layout. We then build ResidualBlocks, which mirrors a bottleneck ResNet-V2 design (He et al., 2016): a $3 \times 3 \times 3$ conv–LN–GELU (Hendrycks & Gimpel, 2023) stem, a $1 \times 1 \times 1 \rightarrow 3 \times 3 \times 3$ bottleneck branch that doubles the channel count, and two skip paths: (i) projection shortcut for stride/width mismatches and (ii) "within-block" residual that re-adds the pre-bottleneck activation just before the final GELU. The encoder begins with a 96-channel input, expands to 192 channels, and applies two strided ResidualBlock units to down-sample from $40^3$ latents to $20^3$ and then $10^3$. Thus we have a narrow bottleneck of 24 channels that serves as the latent code. The symmetric decoder inverts this pathway with a $3 \times 3 \times 3$ convolution, residual processing, and two transposed-conv up-sampling stages that restore the original resolution, all regularised by spatial Dropout in 3D. U-Net–style (Ronneberger et al., 2015) skip connections add encoder feature maps to decoder activations whenever spatial dimensions match.

## A.2 TRAINING

Our results are collected on an Intel(R) Core(TM) i7-13700K CPU machine with one NVIDIA RTX4090 GPU and 128GB memory. We list the choices we make during training for each encoder-generator pair.

For all the models we train with MeshAnything (Chen et al., 2024b) as the encoder, we use the 350 million parameter version of MeshAnything and use the features created by MeshAnything as the encoded inputs. These encoded inputs (`257, 1024`) are relatively larger compared to our desired compression size.

**MeshAnything (Chen et al., 2024b)-InstantMesh (Xu et al., 2024).** While generating the dataset, we ensure that the input condition images are in the RGBA format and have no background. We experiment to have the mapping networks generate: multiview ViT embeddings (Dosovitskiy et al., 2021) that InstantMesh uses as well as the triplane representation InstantMesh uses. We experimentally observed better performance with generating triplane representations, thus our results report this setting.

**MeshAnything (Chen et al., 2024b)-Open LRM (Hong et al., 2024; He & Wang, 2023).** While generating the dataset, we ensure that the input condition images are in the RGBA format and have no background. Due to the lack of public code for the LRM, we use the OpenLRM implementation. We experiment with both the `openlrm-mix-base-1.1` and `openlrm-obj-base-1.1`, and we experimentally observed the `openlrm-obj-base-1.1` to have better performance. We train the mapping networks to generate the triplane latents for LRM.

**MeshAnything (Chen et al., 2024b)-Shap-E (Jun & Nichol, 2023).** We experiment with Shap-E in the text-to-image mode. The mapping networks generate the implicit MLP representations.

**PointNet++ (Qi et al., 2017)-LION (Zeng et al., 2022).** We experiment with the `SSG` and `MSG` versions of PointNet++ and experimentally observed `SSG` without normals to work the best. We use the PointNet++ checkpoints pre-trained for classification. We consider the outputs from the point set feature learning module as the encoded representations. We use the `all` categories model for LION. The mapping networks are trained to generate both the global and local latents for LION.

We provide the model sizes in Tb. 5. We provide the hyperparameters we use to train the mapping networks in Tb. 7. We provide the sizes of latent spaces in Tb. 6.

## B EVALUATION DETAILS

Squeeze3D leverages pre-trained generative priors strictly as a decoding mechanism for compressing arbitrary 3D data, distinct from compressing the native outputs of generative models. As such, direct quantitative comparisons with the original generative backbones are effectively precluded, given that these models condition on differing modalities (e.g., text or images) rather than the geometric inputs required for the compression task.

### B.1 BASELINES

**Meshes.** We compare Squeeze3D against non-learned baselines: Draco (Galligan et al., 2018) (with multiple settings) and Corto (Lab, 2025), the best performing mesh compression techniques. For all settings of Draco, we use a compression level of 10. The setting $\S$ represents 14 bits for the position attribute, 14 bits for texture coordinates, 14 bits for normal vector attributes, and 14 bits for any generic attribute. The setting $\ddagger$ represents 11 bits for the position attribute, 10 bits for texture coordinates, 8 bits for normal vector attributes, and 8 bits for any generic attribute. The setting $\dagger$ represents 7 bits for the position attribute, 7 bits for texture coordinates, 7 bits for normal vector attributes, and 7 bits for any generic attribute. The setting $*$ represents 4 bits for the position attribute, 4 bits for texture coordinates, 4 bits for normal vector attributes, and 4 bits for any generic attribute. We also compare our approach against learned approaches: Neural Subdivision (Liu et al., 2020a), DeepSDF (Park et al., 2019), and Neural Geometry Fields (Edavamadathil Sivaram et al., 2024). All

Table 7: **Training Hyperparameters.** We show the training hyperparameters for the mapping networks we train.

| Hyperparameter | LRM | Shap-E | InstantMesh | NeRF-MAE |
|---|---|---|---|---|
| Training Precision | FP-32 | FP-32 | FP-32 | FP-32 |
| Compressed Size | 1024 | 1024 | 770 | 24000 |
| Dropout | 0.35 | 0.35 | 0.35 | 0.2 |
| Epochs | 700 | 700 | 700 | 2000 |
| Batch Size | 16 | 8 | 16 | 4 |
| Optimizer | Muon | Adam | Muon | Muon |
| Optimizer Parameters | NS = 6 $\beta = 0.95$ | $\lambda = 10^{-2}$ $\beta_1 = 0.9$ $\beta_2 = 0.999$ | NS = 6 $\beta = 0.95$ | NS = 6 $\beta = 0.95$ |
| Initial Learning Rate | $10^{-2}$ | $10^{-4}$ | $10^{-3}$ | $10^{-2}$ |
| Final Learning Rate | $10^{-7}$ | $10^{-4}$ | $10^{-7}$ | $10^{-5}$ |
| Scheduler | Linear Decay | Constant | Linear Decay | Linear Decay |
| Epochs Decay | 700 | N/A | 600 | 1000 |
| Gradient Accum. Steps | 1 | 2 | 1 | 1 |
| Gradient Clipping | None | None | None | None |
| Hyperparameter | LION (1024) | LION (2048) | LION (4096) | LION (8192) |
| Training Precision | FP-32 | FP-32 | FP-32 | FP-32 |
| Hidden Size | 1024 | 2048 | 4096 | 8192 |
| Dropout | 0.3 | 0.3 | 0.3 | 0.3 |
| Epochs | 4000 | 4000 | 1000 | 400 |
| Batch Size | 16 | 16 | 16 | 16 |
| Optimizer | Muon | Muon | Muon | Muon |
| Optimizer Parameters | $\beta = 0.95$ NS = 6 | $\beta = 0.95$ NS = 6 | $\beta = 0.95$ NS = 6 | $\beta = 0.95$ NS = 6 |
| Initial Learning Rate | $10^{-3}$ | $10^{-3}$ | $10^{-3}$ | $10^{-3}$ |
| Final Learning Rate | $10^{-7}$ | $10^{-7}$ | $10^{-7}$ | $10^{-7}$ |
| Scheduler | Linear Decay | Linear Decay | Linear Decay | Linear Decay |
| Epochs Decay | 1000 | 1000 | 1000 | 1000 |
| Gradient Accum. Steps | 1 | 1 | 1 | 1 |
| Gradient Clipping | None | None | None | None |

the baseline models we compare against do not support texture images, thus we pair these methods with JPEG to compress the accompanying texture images.

**Point Clouds.** We compare Squeeze3D against Draco (Galligan et al., 2018), the best performing point cloud compression technique. For all settings of Draco, we use a compression level of 10. The setting [§] represents 14 bits for the position attribute, 14 bits for texture coordinates, 14 bits for normal vector attributes, and 14 bits for any generic attribute. The setting [‡] represents 11 bits for the position attribute, 10 bits for texture coordinates, 8 bits for normal vector attributes, and 8 bits for any generic attribute.

**Radiance Fields.** We compare Squeeze3D against SparsePCGC (Wang et al., 2023), and VQRF (Li et al., 2023a), the best performing applicable techniques to radiance field compression. Particularly, note that our method compresses arbitrary radiance fields and thus most existing NeRF and 3D Gaussian Splat compression echniques are not applicable to our problem. To evaluate VQRF (Li et al., 2023a), we use the voxel pruning and vector quantization steps which are relevant for compressing radiance field grids.

Table 8: **Mesh Compression.** Quantitative comparison of Squeeze3D with state-of-the-art 3D mesh compression methods, including geometry-based metrics Chamfer Distance and EMD. We also report the compression and decompression wall-clock times.

| Method | CR (×) (MB) ↑ | Compress (ms) ↓ | Decompress (ms) ↓ | PSNR ↑ | MS-SSIM ↑ | LPIPS ↓ | Chamfer ↓ | EMD ↓ |
|---|---|---|---|---|---|---|---|---|
| Draco* (Galligan et al., 2018) + JPEG | 6.92 (6.43 / 0.93) | 70.30 | 29.97 | 15.45 (±1.27) | 0.7468 (±0.09) | 0.2437 (±0.09) | 1.2746 | 0.0544 |
| Draco† (Galligan et al., 2018) + JPEG | 6.70 (6.43 / 0.96) | 69.05 | 28.33 | 23.33 (±1.32) | 0.9576 (±0.03) | 0.1039 (±0.06) | 0.8739 | 0.0385 |
| Draco‡ (Galligan et al., 2018) + JPEG | 6.61 (6.43 / 0.97) | 68.15 | 27.46 | 38.91 (±1.30) | 0.9992 (±0.00) | 0.0045 (±0.00) | 0.4214 | 0.0108 |
| Draco§ (Galligan et al., 2018) + JPEG | 6.20 (6.43 / 1.04) | 66.30 | 25.81 | 48.55 (±1.54) | 1.0000 (±0.00) | 0.0004 (±0.00) | 0.2060 | 0.0044 |
| Corto (Lab, 2025) + JPEG | 45.93 (6.43 / 0.14) | 50.52 | 8.20 | 20.92 (±2.79) | 0.8619 (±0.08) | 0.1374 (±0.08) | 0.9644 | 0.0438 |
| Neural Subd. (Liu et al., 2020a) + JPEG | 11.28 (6.43 / 0.57) | 61104.12 | 0.00 | 15.95 (±2.18) | 0.8525 (±0.04) | 0.1513 (±0.05) | 1.1125 | 0.0496 |
| DeepSDF (Park et al., 2019) + JPEG | 131.22 (6.43 / 0.05) | 887.78 | 578.53 | 8.47 (±0.23) | 0.7039 (±0.07) | 0.3704 (±0.08) | 1.5842 | 0.0617 |
| NGF (Edavamadathil Sivaram et al., 2024) + JPEG | 42.87 (6.43 / 0.15) | 152637.87 | 507.21 | 35.45 (±3.02) | 0.9987 (±0.08) | 0.0054 (±0.03) | 0.3866 | 0.0130 |
| Squeeze3D (InstantMesh) | 2187.07 (6.43 / 0.003) | 270.24 | 1476.00 | 27.50 (±3.13) | 0.9796 (±0.02) | 0.0274 (±0.02) | 0.3930 | 0.0084 |

Table 9: **Radiance Field Results.** Quantitative comparison of Squeeze3D for compressing radiance fields from the ai subset of the NeRF-MAE dataset (Appendix B.3).

| Method | CR (×) (MB) ↑ | Compress (ms) ↓ | Decompress (ms) ↓ | PSNR ↑ | MS-SSIM ↑ | LPIPS ↓ |
|---|---|---|---|---|---|---|
| Ours | 657.89 (59.21 / 0.09) | 45.63 | 75.80 | 22.40 | 0.8958 | 0.1400 |

## B.2 COLLECTION OF ADDITIONAL MESHES FOR EVALUATION

Since Squeeze3D is designed as a compression-decompression framework that should not require per-scene retraining, the generalization capability of our mapping networks to unseen 3D models is very important. Thus, we also collected a very diverse dataset of 500 high-quality 3D meshes from the following sources,

- OpenLRM Demo: `https://huggingface.co/spaces/zxhezexin/OpenLRM`
- InstantMesh Demo: `https://huggingface.co/spaces/TencentARC/InstantMesh`
- Hunyuan3D-2 Demo: `https://huggingface.co/spaces/tencent/Hunyuan3D-2`
- TRELLIS Demo: `https://huggingface.co/spaces/JeffreyXiang/TRELLIS`
- SPAR3D Demo: `https://huggingface.co/spaces/stabilityai/stable-point-aware-3d`
- SORA-3D Demo: `https://huggingface.co/spaces/ginipick/SORA-3D`
- A subset of ABO (Collins et al., 2022) composed of 342 meshes

## B.3 COLLECTION OF ADDITIONAL RADIANCE FIELDS FOR EVALUATION

Since Squeeze3D is designed as a compression-decompression framework that should not require per-scene retraining, the generalization capability of our mapping networks to unseen 3D models is very important. Thus, we also report results on the unseen "ai" subset of the NeRF-MAE dataset (Irshad et al., 2024).

## C ADDITIONAL RESULTS

### C.1 ADDITIONAL RESULTS FOR MESH COMPRESSION

We show the results for mesh compression on our test set or the $\mathcal{T}$ set with 3D metrics (Chamfer distance and EMD) as well as wall-clock times for compression and decompression in Tb. 8.

### C.2 ADDITIONAL RESULTS FOR POINT CLOUD COMPRESSION

We show the results for point cloud compression on our test set with 3D metrics (Chamfer distance and EMD) in Tb. 10.

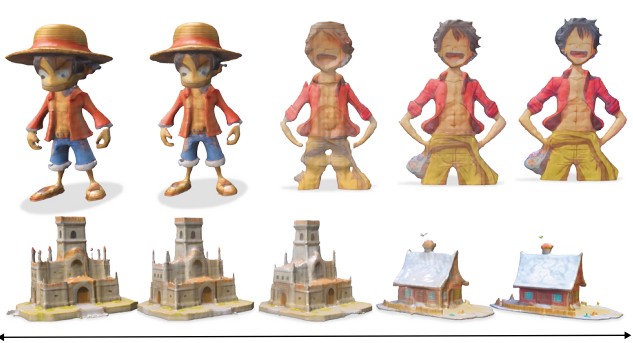

Linear Interpolation

Figure 8: **Interpolation.** The compressed representations we obtain can also be interpolated. In these examples we obtain the compressed representation for the leftmost and rightmost meshes and linearly interpolate between them.

Table 10: **Point Cloud Compression with Geometry Metrics.** Quantitative comparison of Squeeze3D with state-of-the-art point cloud compression methods, including geometry-based metrics Chamfer Distance and EMD.

| Method | CR (×) (KB) ↑ | Compress (ms) ↓ | Decompress (ms) ↓ | PCQM ↑ | PointSSIM ↑ | Chamfer ↓ | EMD ↓ |
|---|---|---|---|---|---|---|---|
| Draco[‡] (Galligan et al., 2018) | 15.64 (117 / 7.48) | 0.35 | 0.34 | 3.2875 (± 0.13) | 0.9722 (± 0.11) | 0.0285 | 0.0062 |
| Draco[§] (Galligan et al., 2018) | 22.41 (117 / 5.22) | 0.25 | 0.23 | 2.1039 (± 0.08) | 0.9535 (± 0.12) | 0.0419 | 0.0095 |
| Squeeze3D (LION) | 58.50 (117 / 2.00) | 3.85 | 12.74 | 1.8437 (± 1.13) | 0.4484 (± 0.11) | 0.0285 | 0.0062 |

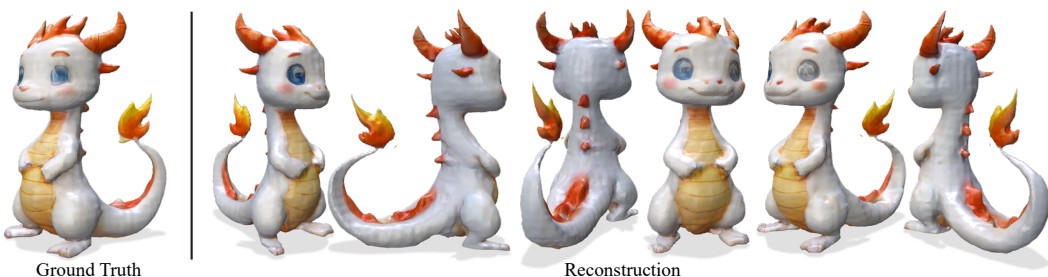

Ground Truth      Reconstruction

Figure 9: **Multi-view visualization of compressed and reconstructed meshes.** The consistent appearance across different viewing angles demonstrates that Squeeze3D learns correct transformations between latent spaces and produces coherent 3D reconstructions. This confirms that our compressed representation encodes complete 3D information rather than view-dependent features.

### C.3 ADDITIONAL RESULTS FOR RADIANCE FIELD COMPRESSION

Our mapping networks are trained on the dataset we derived from NeRF-MAE (Irshad et al., 2024), but we test Squeeze3D on only the radiance fields in the `ai` subset of the dataset which was collected by seprately than other datsets. On our collection of meshes, Squeeze3D achieves on average only a 4.22 dB reduction in the PSNR as we show in Tb. 9.

### C.4 QUALITATIVE RESULTS ON MESHES.

We also notice that our approach works well with high-resolution complex meshes as we show in Fig. 12. In Fig. 9, we show a mesh we reconstruct with our method through multiple camera angles to demonstrate that our approach learns a correct transformation between the latent spaces and the reconstruction is consistent. We also observe that the compression representation our method learns can also be interpolated as we show in Fig. 8. In Fig. 11 we show examples using two different generative models indicating that Squeeze3D can be extended to use other generative models for compression. In Fig. 10, we observe that our method learns to effectively represent intricate geometrical details.

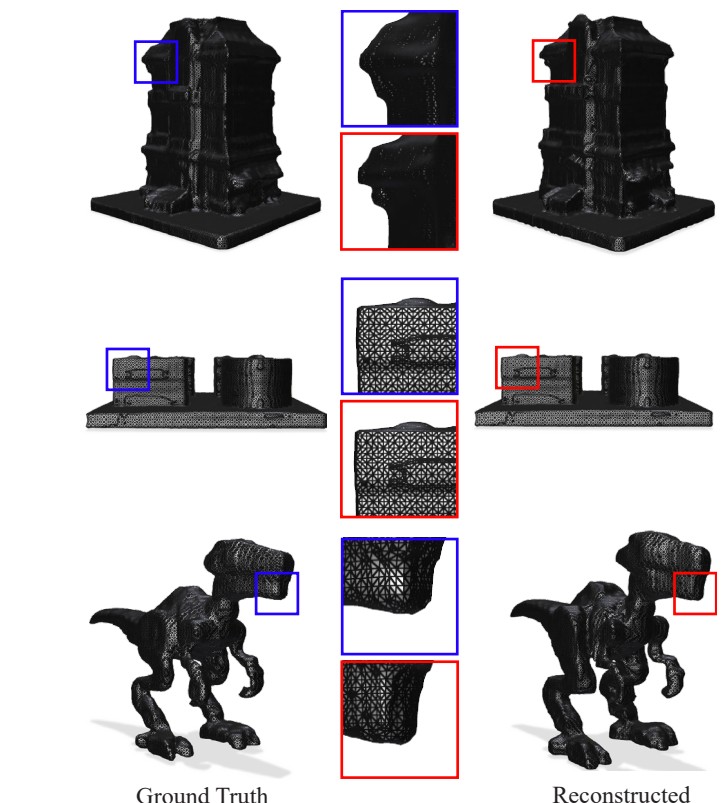

Ground Truth                                      Reconstructed

Figure 10: **Squeeze3D preserves geometry details.** We show some meshes compressed with Squeeze3D as wireframes. Notice that Squeeze3D preserves many finegrained geometric details.

Table 11: **Ablations.** Ablation study on compressed representation size. We analyze how varying the dimensionality of the compressed latent space affects compression time, decompression time, and reconstruction quality (measured by PCQM and PointSSIM).

| Size ($\mathbf{z}_{comp}$) | Compression (s) | Decompression (s) | PCQM ↑ | PointSSIM ↑ |
|---|---|---|---|---|
| 1024 | 3.44 | 12.33 | 1.4047 | 0.3640 |
| 2048 | 3.51 | 12.39 | 1.3311 | 0.4249 |
| 4096 | 3.85 | 12.74 | 1.8437 | 0.4318 |
| 8192 | 5.30 | 14.19 | 1.5665 | 0.4473 |

## C.5 ABLATIONS

We conduct several ablation studies to better understand the behavior of Squeeze3D and analyze the trade-offs between compression ratio, reconstruction quality, and computational efficiency. First, we investigate how varying the size of our compressed representation affects decompression time. In Tb. 11, we show that the compressed size affects the compression and decompression times proportionally since changes in the compressed size affect the size of the neural network. Next, we examine how different compression sizes affect reconstruction quality. Tb. 11 shows a correlation between latent dimension and reconstruction quality. We observe substantial improvements in PointSSIM when increasing the compressed size from 1024 to 2048, however, this improvement in quality starts diminishing beyond the compressed size of 2048. We perform an ablation of training with and without gram loss (§3.2) in Tb. 12.

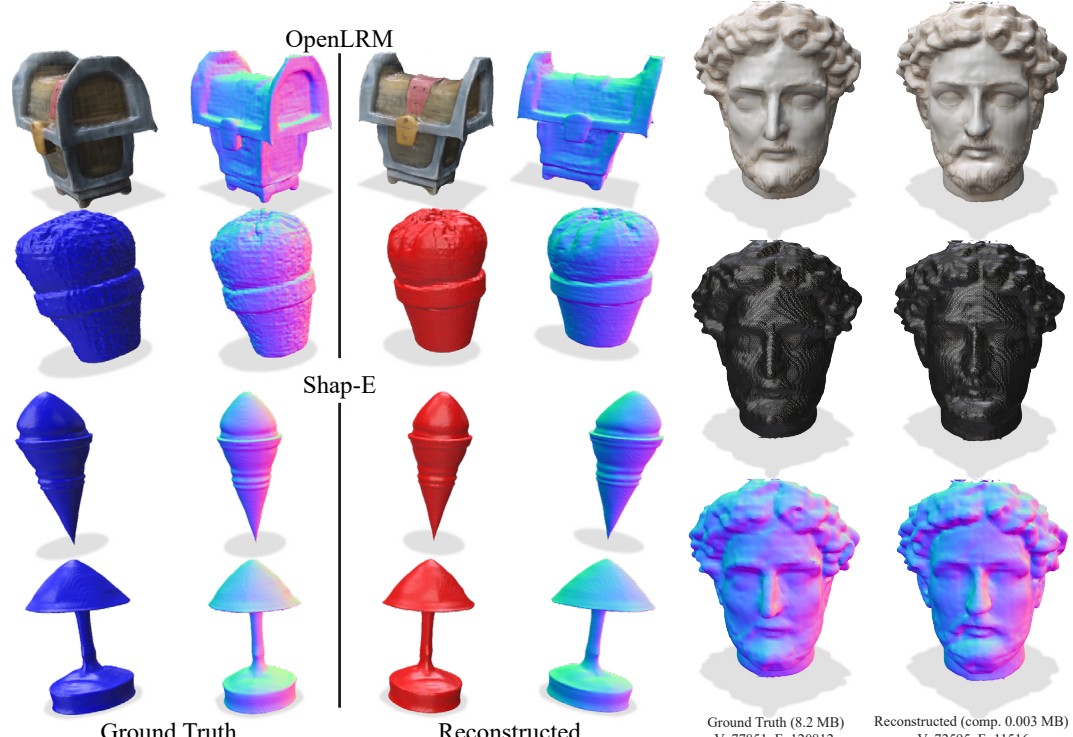

Figure 11: **Compression results using different 3D generators.** Squeeze3D is agnostic to the choice of a 3D generation model. Thus, we show compression results with the 3D generators: OpenLRM, and Shap-E. We choose 3D meshes that lie in the representation capacity of the chosen 3D generators.

Figure 12: **Compressing Complex Meshes.** Squeeze3D can be used to compress highly complex textured 3D meshes (in this case 77851 vertices and 120812 faces).

Table 12: **Ablation.** Quantitative comparison of Squeeze3D (InstantMesh) with and without Gram-loss.

| Method | CR (×) (MB) ↑ | Compress (ms) ↓ | Decompress (ms) ↓ | PSNR ↑ | MS-SSIM ↑ | LPIPS ↓ |
|---|---|---|---|---|---|---|
| Squeeze3D (InstantMesh) | 2187.07 (6.43 / 0.003) | 270.24 | 1476.00 | 27.50 (±3.13) | 0.9796 (±0.02) | 0.0274 (±0.02) |
| w/o Gram-loss | 2187.07 (6.43 / 0.003) | 270.24 | 1476.00 | 24.20 (±3.50) | 0.9205 (±0.04) | 0.1321 (±0.035) |

## C.6 COMPARISON WITH NATIVE GENERATIVE MODEL ENCODERS

We perform an additional experiment to compare Squeeze3D against a baseline of using the native encoder and decoder of the generative models themselves for compression. This baseline effectively uses the latent space of the generative model as the compressed representation. Since these generative models are designed for high-fidelity generation rather than compression, their latent spaces are typically high-dimensional.

We perform this comparison for Shap-E (Jun & Nichol, 2023), LION (Zeng et al., 2022), and NeRF-MAE (Irshad et al., 2024). We do not include InstantMesh (Xu et al., 2024) or LRM (Hong et al., 2024) in this comparison because these are feed-forward image-to-3D models that utilize 2D image encoders (e.g., DINO). They lack a native 3D-to-3D encoder that can map a 3D mesh directly to their latent space, making them unsuitable for this specific 3D compression baseline without introducing a lossy intermediate rendering step.

We report the results in Tb. 13 to 15. We observe that the native baselines generally achieve high reconstruction quality (serving as an upper bound) but at significantly lower compression ratios. In contrast, Squeeze3D achieves orders of magnitude higher compression ratios while maintaining comparable visual quality. In some cases, such as with Shap-E, our method can even slightly outperform the native baseline on certain metrics.

Table 13: **Shap-E Comparison.** Comparison between Squeeze3D (MeshAnything $\rightarrow$ Shap-E) and the native Shap-E encoder-decoder.

| Method | CR ($\times$) $\uparrow$ | PSNR $\uparrow$ | MS-SSIM $\uparrow$ | LPIPS $\downarrow$ |
|---|---|---|---|---|
| Shap-E Native Baseline | 1.61 | 25.80 | 0.9600 | 0.0410 |
| Squeeze3D (Shap-E) | 1607.50 | 25.92 | 0.9615 | 0.0398 |

Table 14: **LION Comparison.** Comparison between Squeeze3D (PointNet++ $\rightarrow$ LION) and the native LION encoder-decoder.

| Method | CR ($\times$) $\uparrow$ | PCQM $\uparrow$ | PointSSIM $\uparrow$ | Chamfer ($10^{-3}$) $\downarrow$ |
|---|---|---|---|---|
| LION Native Baseline | 3.60 | 2.1000 | 0.5100 | 0.0210 |
| Squeeze3D (LION) | 58.50 | 1.8437 | 0.4484 | 0.0285 |

## C.7 COMPRESSION RATIO AND RECONSTRUCTION QUALITY TRADE-OFF

To better understand the trade-off between compression ratio and reconstruction quality, we analyze the performance of Squeeze3D against Draco (Galligan et al., 2018) across different compression settings. While many neural compression methods do not offer tunable compression ratios, Squeeze3D allows for different compression levels by varying the size of the compressed latent representation. Similarly, Draco offers tunable compression through its quantization parameters. In Fig. 13, we plot the Pareto frontier for both methods, showing the relationship between compression ratio (x-axis) and reconstruction quality (y-axis, measured by LPIPS). We observe that Draco (Galligan et al., 2018) can only achieve compression ratios which are orders of magnitude lower than Squeeze3D, and as its compressions ratio increases its quality degrades significantly. In contrast, Squeeze3D maintains high visual quality.

## C.8 FAILURE CASES

To better understand the limitations of our approach, we analyze failure cases where Squeeze3D struggles to faithfully reconstruct the input geometry. The primary cause of failure stems from the dependency on the generator's prior distribution. When the input geometry significantly deviates from the distribution of shapes the generator was trained on, the generator may fail to produce the correct details, or the mapping network may fail to find a corresponding code in the generator's latent space that represents the input. We illustrate these failure cases in Fig. 14. We primarily observe loss of fine details, where highly intricate structures or text that are not well-represented in the generator's latent space are smoothed out or ignored. This limitation is twofold: (1) The encoder may fail to capture high-frequency details. (2) Even if the encoder captures these details, the mapping network or the generator's latent bottleneck may filter them out.

## D ADDITIONAL RELATED WORKS

While, we are interested in learning based methods for compression, we include classical compression techniques which are only tangentially related to us, for completeness. include approaches that reorder the structure of triangles and faces in the mesh to enable compressed encodings of elements based on their local structure and perform quantization (Deering, 1995; Taubin & Rossignac, 1998; Rossignac, 1999; Touma & Gotsman, 1998; Galligan et al., 2018). Most of these techniques are lossless and thus achieve high fidelity, however, they are inherently limited in their ability to significantly reduce file sizes since they preserve all details of the mesh representation. To overcome the limitations of lossless compression, lossy techniques have emerged as popular alternatives. Geometry simplification methods aim to reduce the number of polygons in a mesh while retaining as much of the original structure as possible (Lescoat et al., 2020; Garland & Heckbert, 1997; Surazhsky & Gotsman, 2003; Szymczak et al., 2002). Extensions of these methods have incorporated surface intrinsic properties, such as the mesh Laplacian, as a basis for simplification (Lescoat et al., 2020) or error metrics (Garland

Table 15: **NeRF-MAE Comparison.** Comparison between Squeeze3D (NeRF-MAE → NeRF-MAE) and the native NeRF-MAE encoder-decoder.

| Method | CR (×) ↑ | PSNR ↑ | MS-SSIM ↑ | LPIPS ↓ |
|---|---|---|---|---|
| NeRF-MAE Native Baseline | 7.44 | 28.10 | 0.9700 | 0.0550 |
| Squeeze3D (NeRF-MAE) | 619.41 | 26.62 | 0.9533 | 0.0743 |

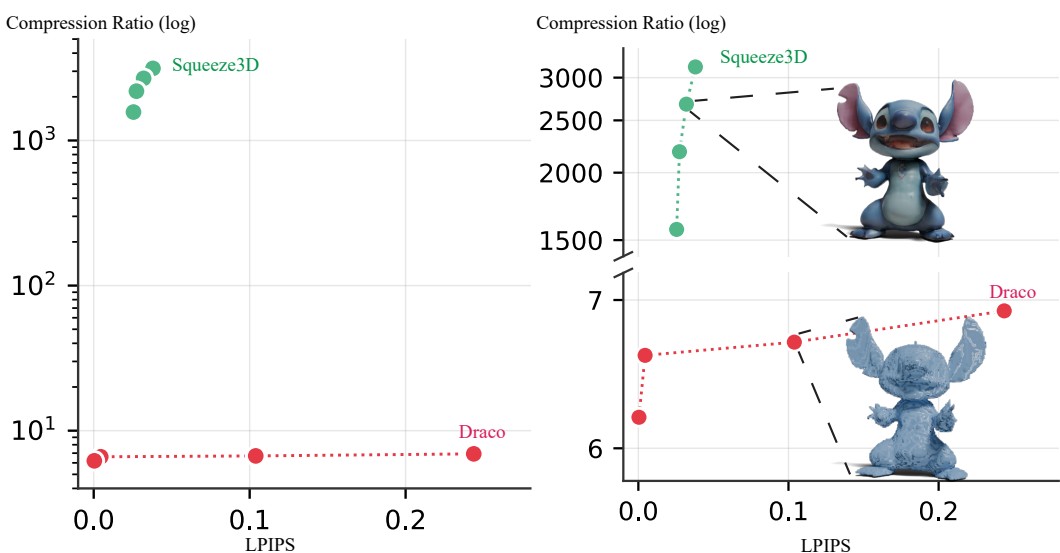

Figure 13: **Compression Ratio and Reconstruction Quality Trade-off.** We show the trade-off between compression ratio and reconstruction quality (LPIPS) for Squeeze3D and Draco. Squeeze3D maintains high quality even at extreme compression ratios, whereas Draco's quality degrades significantly as compression increases.

& Heckbert, 1997; Cohen et al., 1998) and couple this with entropy coding (Lab, 2025). Recently, there have also been approaches that couple these with learned models (Potamias et al., 2022).

Traditional subdivision algorithms (Zorin et al., 1996; Catmull & Clark, 1998; Loop, 1987) refine coarse meshes by splitting polygonal faces into finer elements, often paired with displacement mapping (Hoppe et al., 1994) to enhance detail. However, these methods rely on hard-coded priors and fixed polynomial interpolations, which can overly smooth the reconstructed geometry and fail to capture intricate details. Neural approaches (Hertz et al., 2020; Liu et al., 2020a) address these limitations by embedding geometric information into learnable parameters.

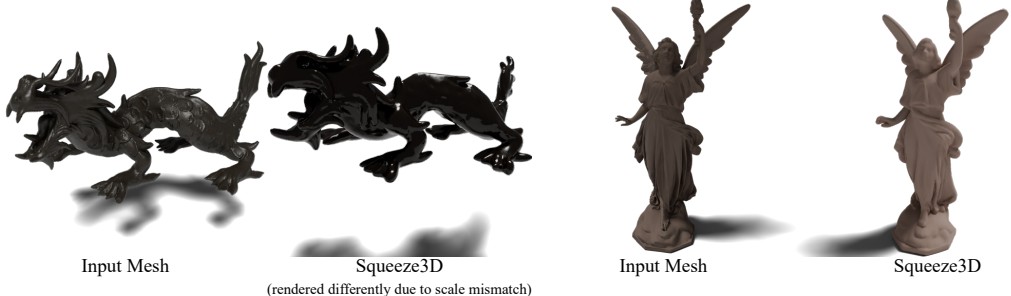

Figure 14: **Failure Cases.** We show examples where Squeeze3D fails to accurately reconstruct the input. (Left) Input mesh with highly intricate details or text. (Right) The reconstruction is smoothed, losing the fine-grained text or surface texture.

# E    SOCIETAL IMPACT

Our method has the potential to advance social good by lowering the bandwidth, storage, and energy requirements associated with the ever-growing volumes of 3D data. By shrinking large meshes, point clouds, and radiance fields to kilobytes, Squeeze3D can make high-fidelity 3D assets practical for mobile devices, distance learning, cultural-heritage archiving, and tele-presence, thereby broadening access to immersive content while also reducing the carbon footprint of cloud infrastructure. However, extreme compression built on powerful generative priors also carries risks: it may facilitate unauthorized replication and distribution of copyrighted 3D models; compressed latents could be used to conceal contraband or embed hidden payloads; and the ease of disseminating photorealistic 3D scenes may accelerate the creation of deceptive "deep-fake" environments.

# F    RESULTS LIBRARY

We present additional results in Fig. 15 to 27.

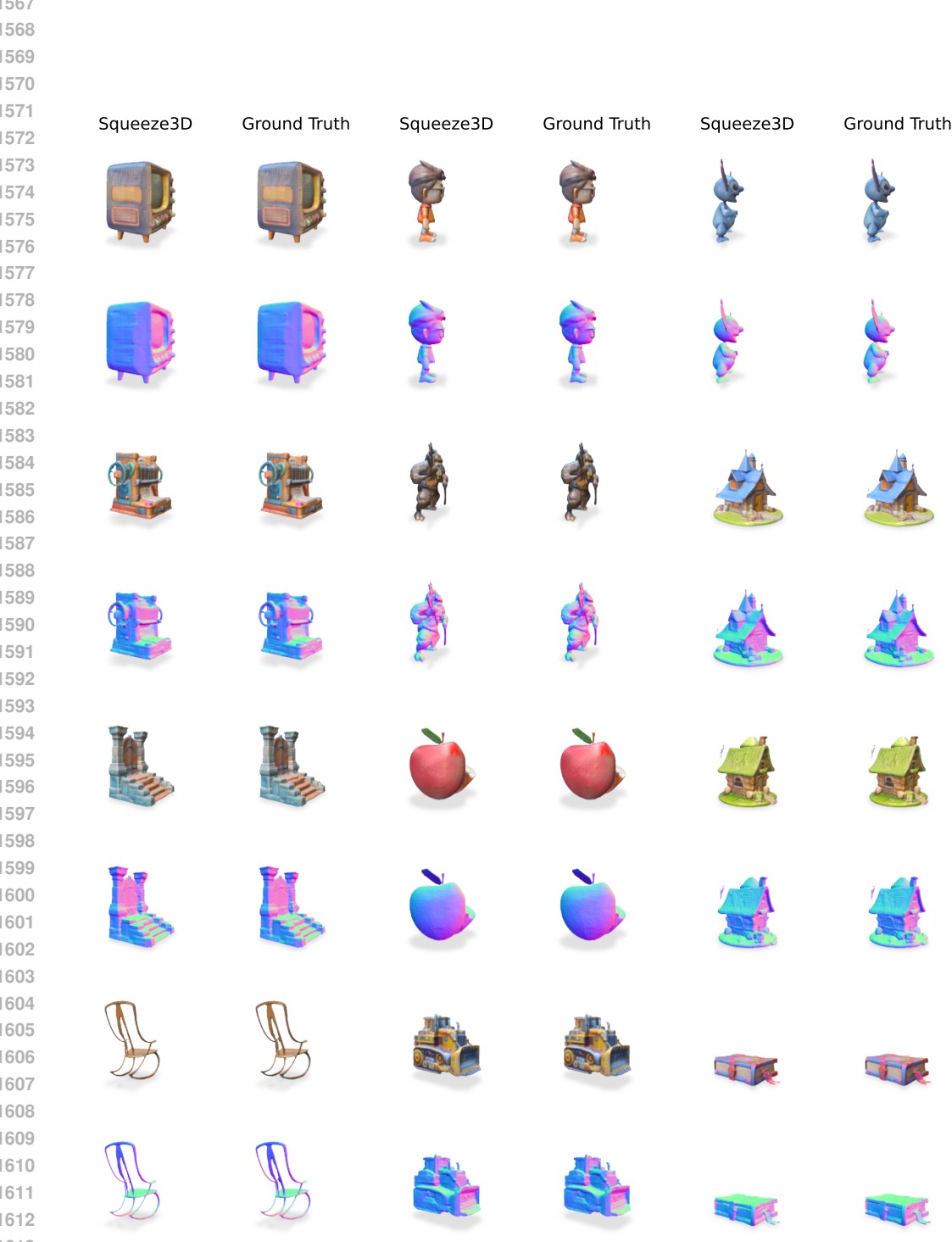

Figure 15: Results Library.

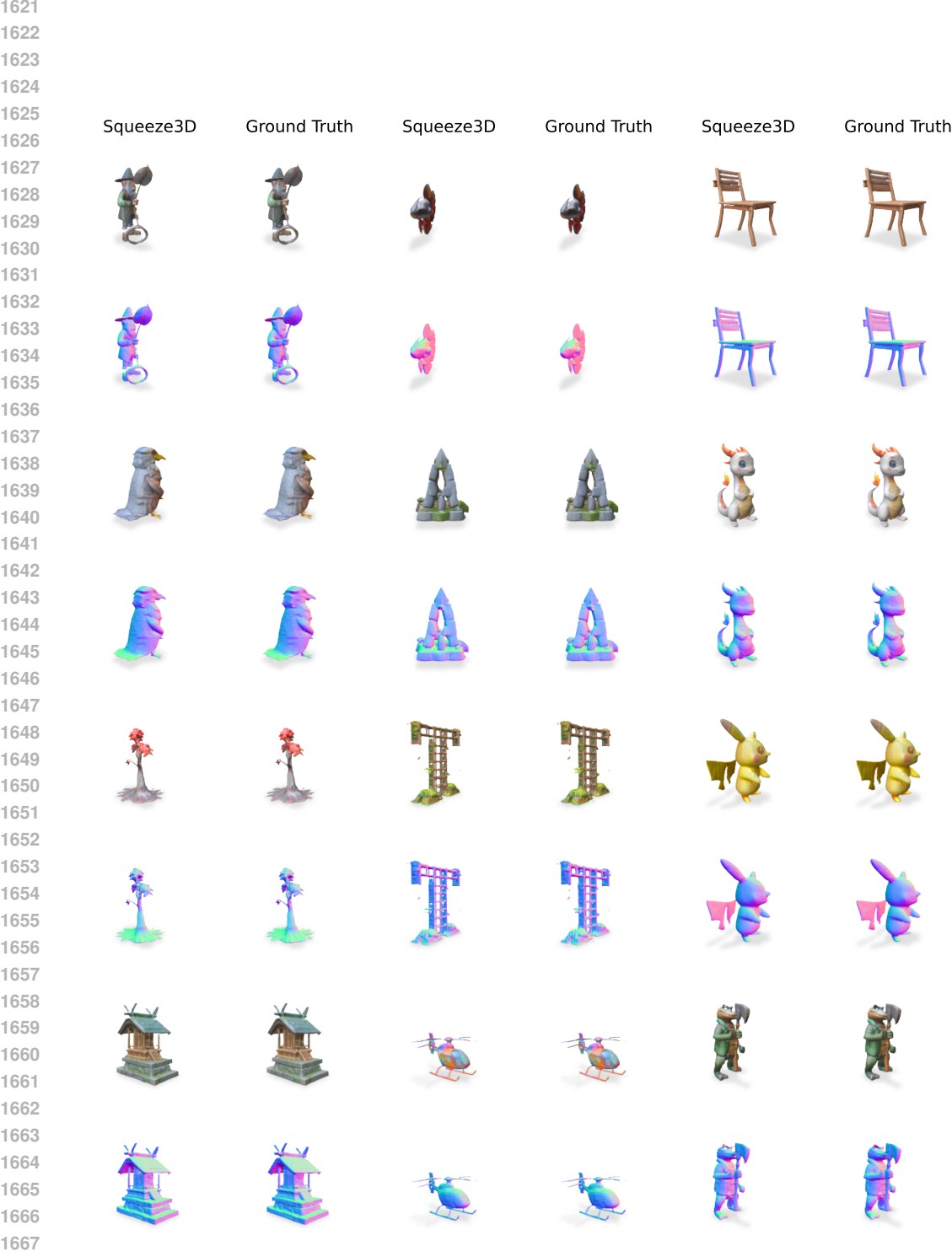

Figure 16: Results Library.

Squeeze3D    Ground Truth    Squeeze3D    Ground Truth    Squeeze3D    Ground Truth

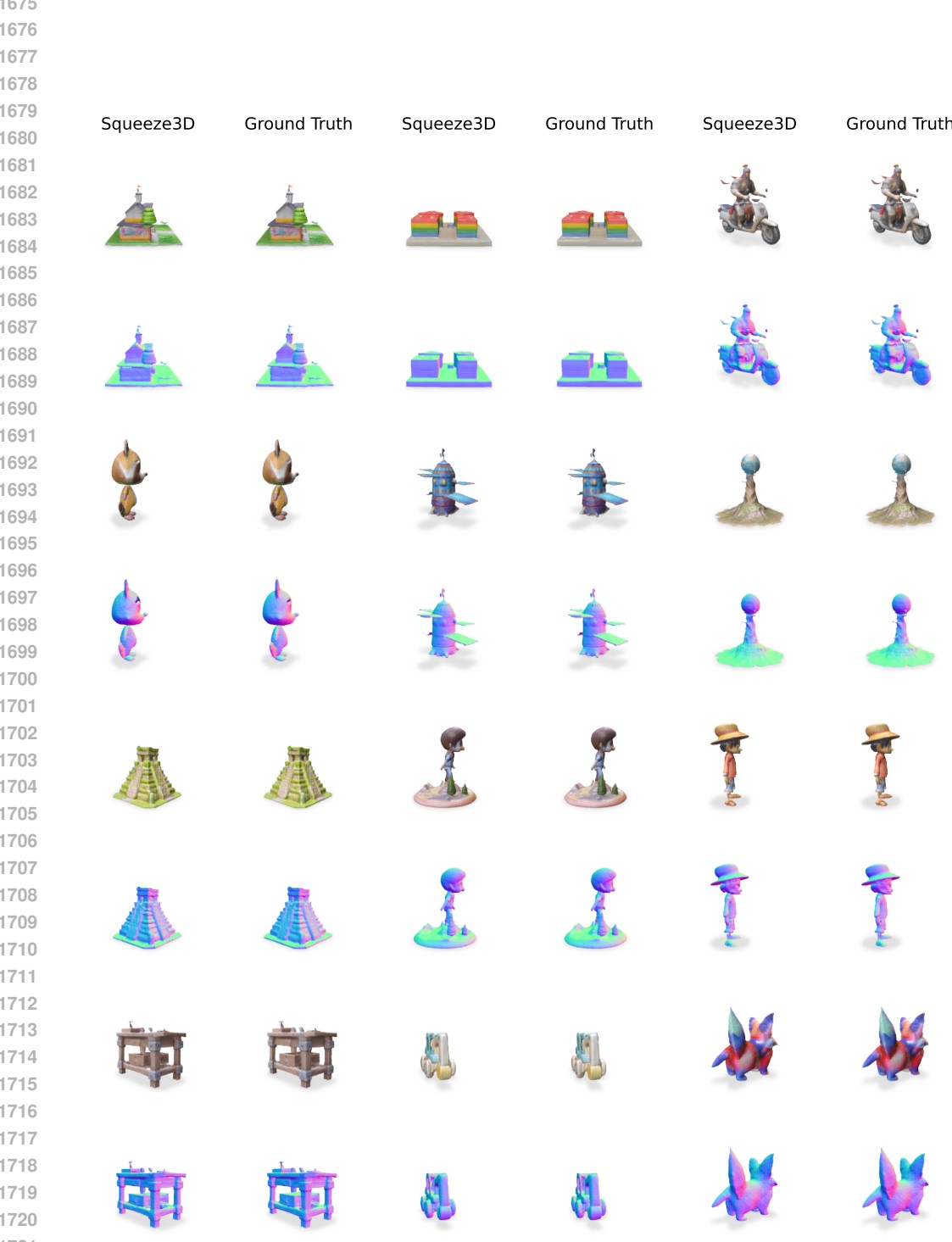

Figure 17: Results Library.

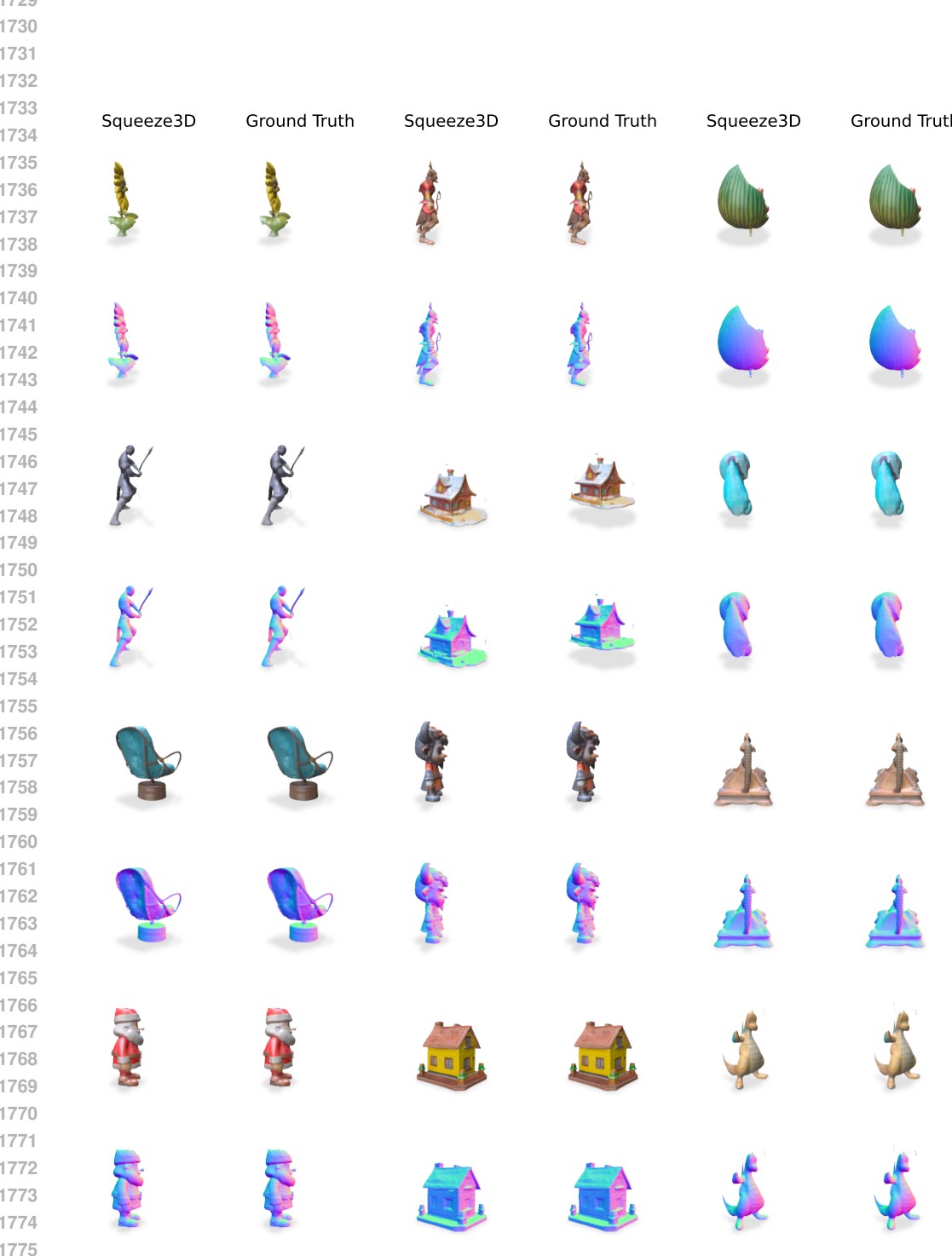

Figure 18: Results Library.

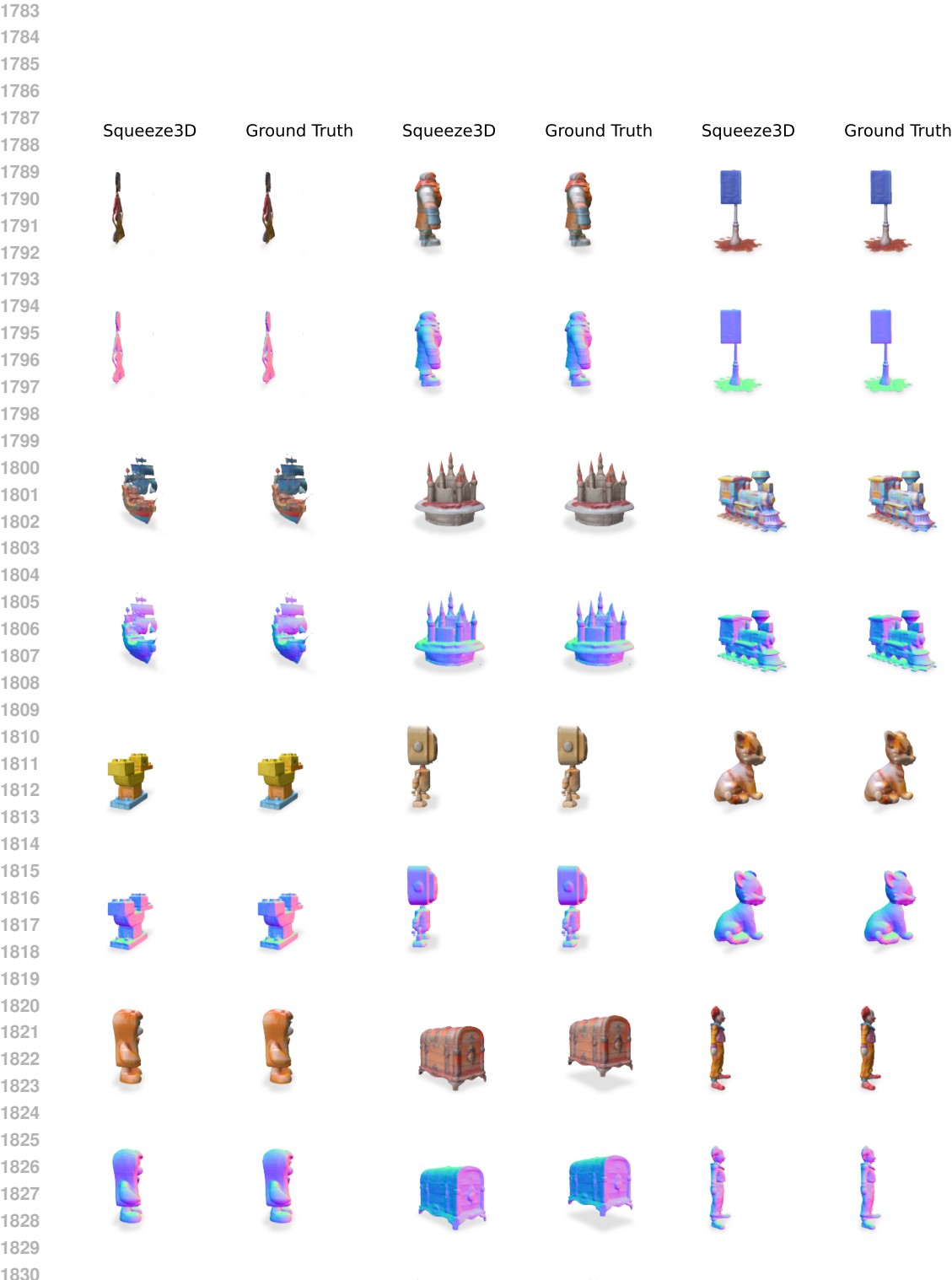

Figure 19: Results Library.

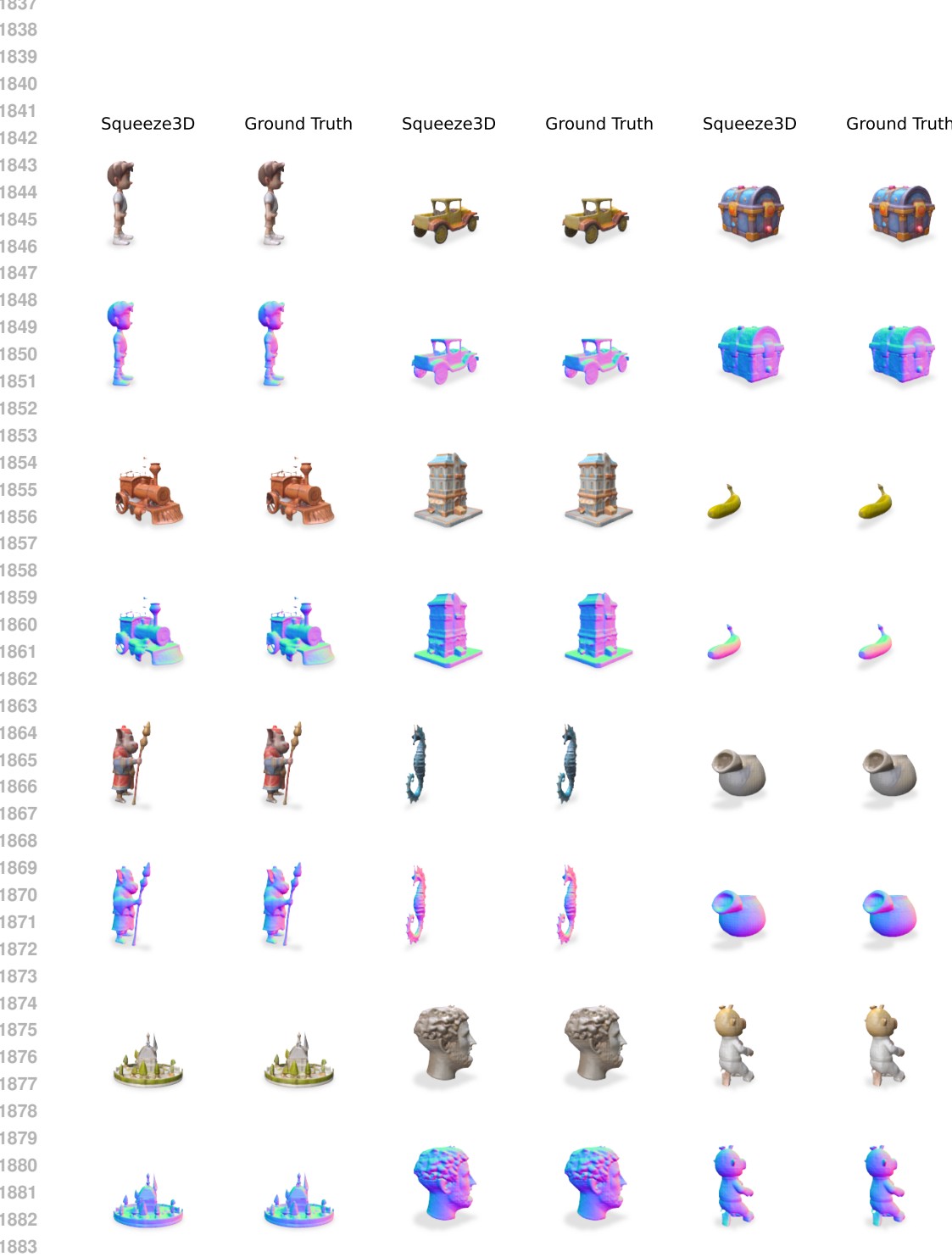

Figure 20: Results Library.

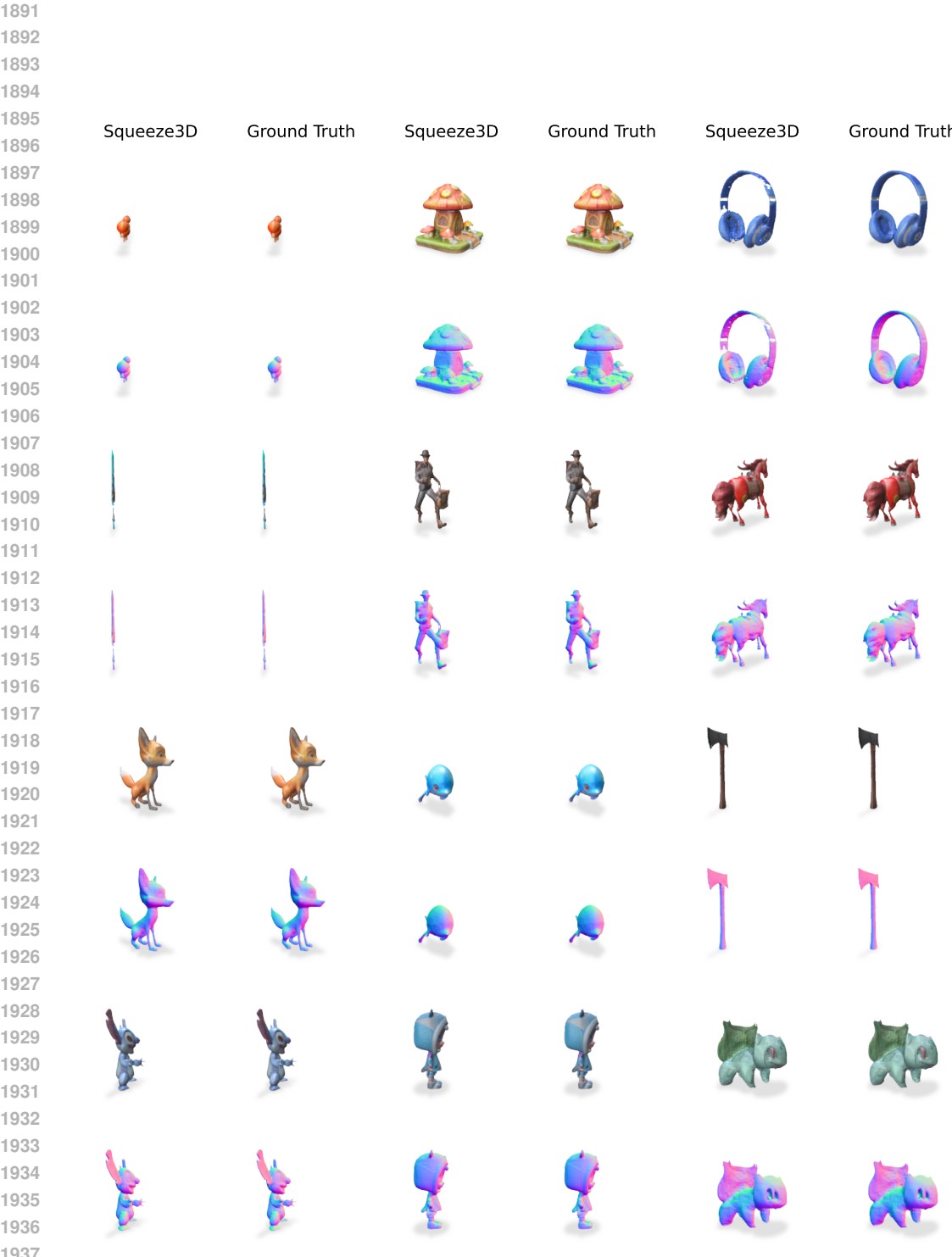

Figure 21: Results Library.

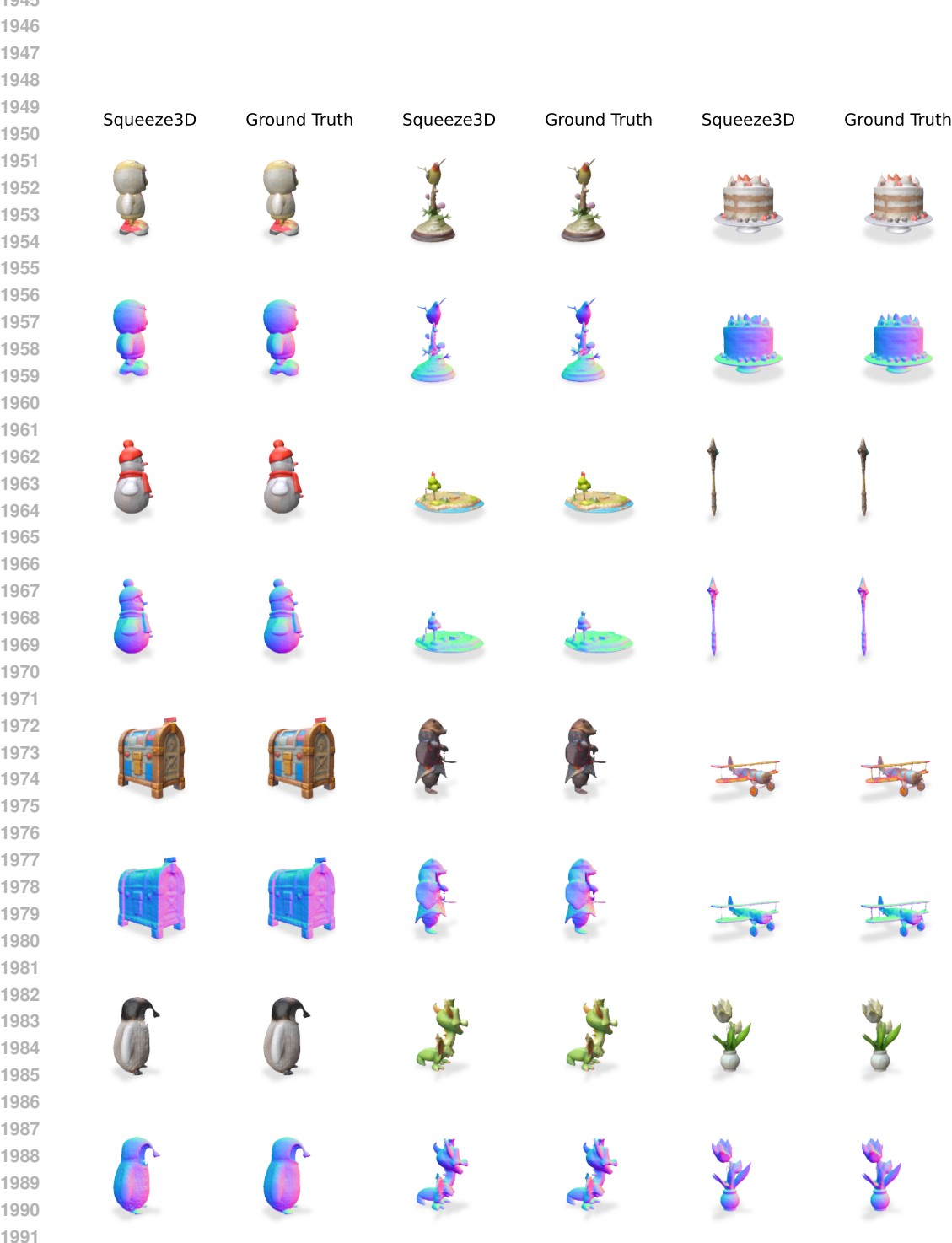

Figure 22: Results Library.

1998
1999
2000
2001
2002
2003
2004
2005
2006
2007
2008
2009
2010
2011
2012
2013
2014
2015
2016
2017
2018
2019
2020
2021
2022
2023
2024
2025
2026
2027
2028
2029
2030
2031
2032
2033
2034
2035
2036
2037
2038
2039
2040
2041
2042
2043
2044
2045
2046
2047
2048
2049
2050
2051

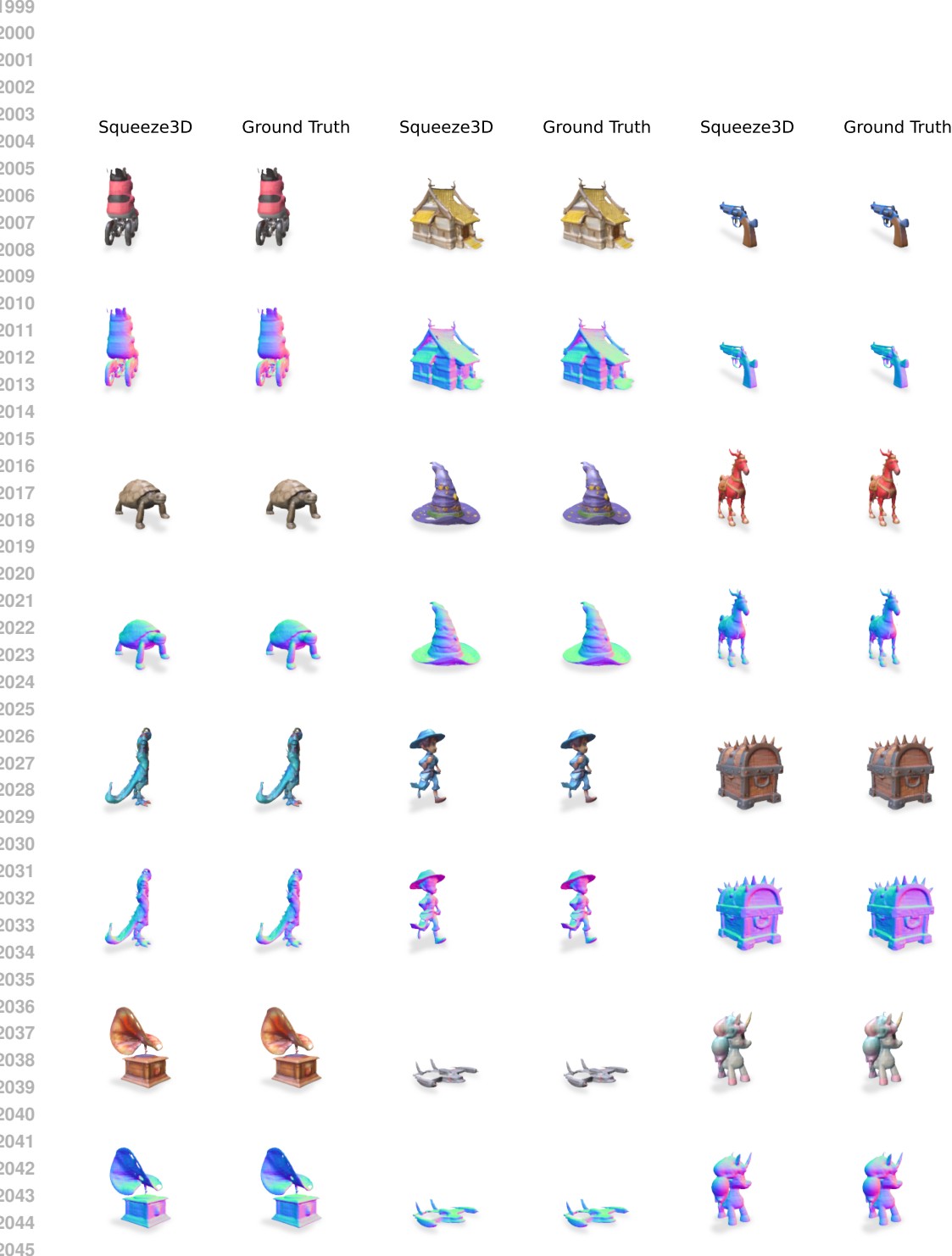

Figure 23: Results Library.

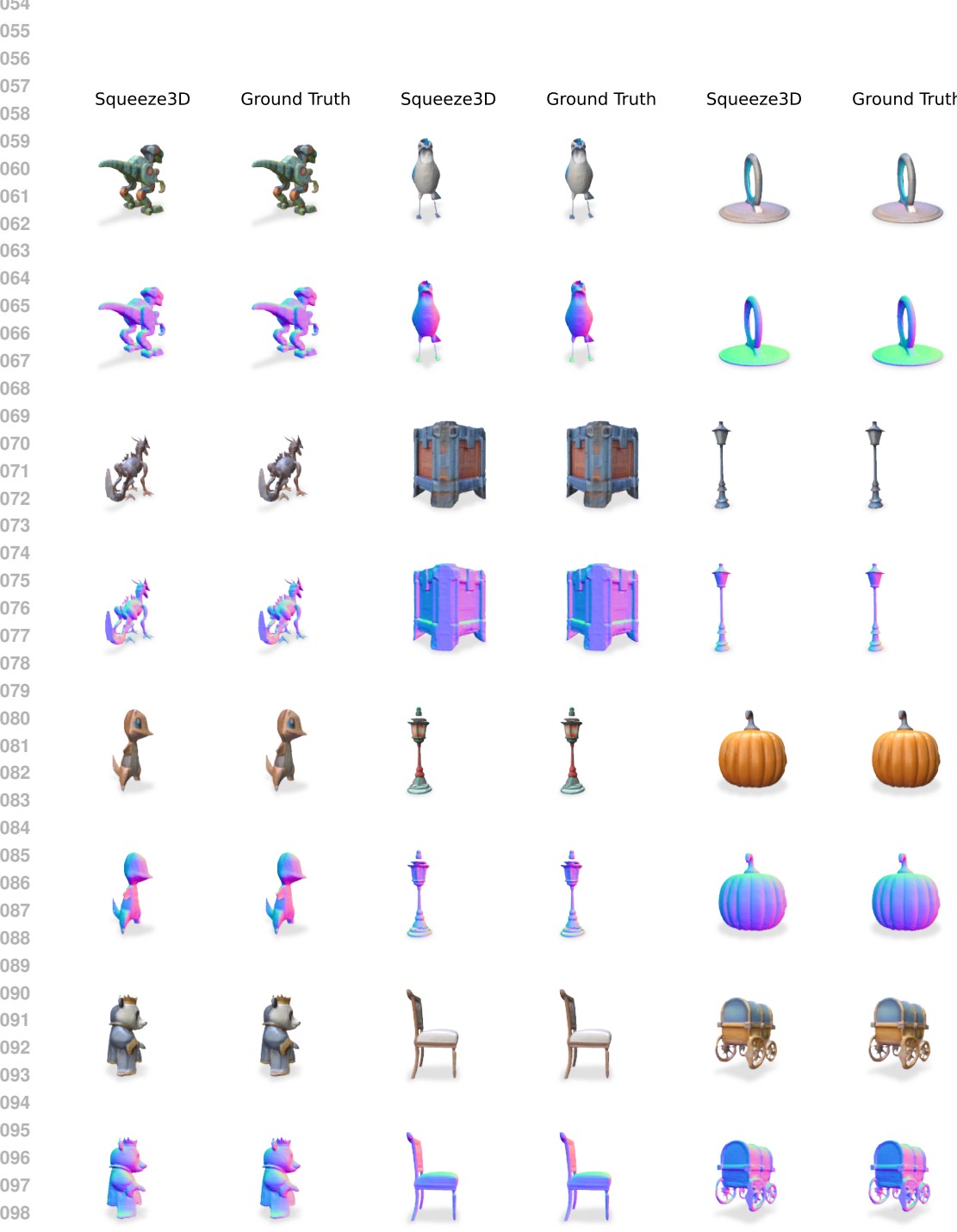

Figure 24: Results Library.

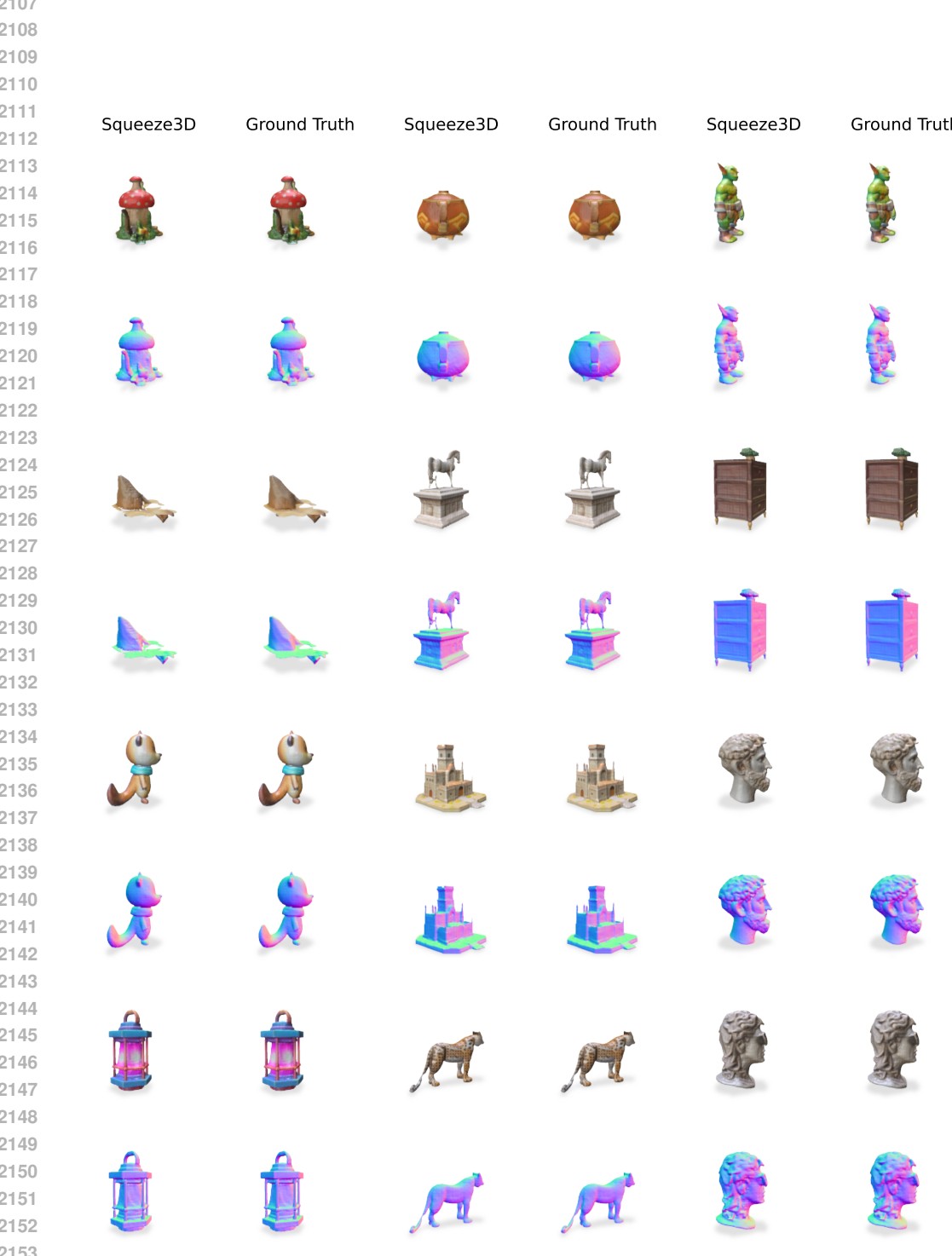

Figure 25: Results Library.

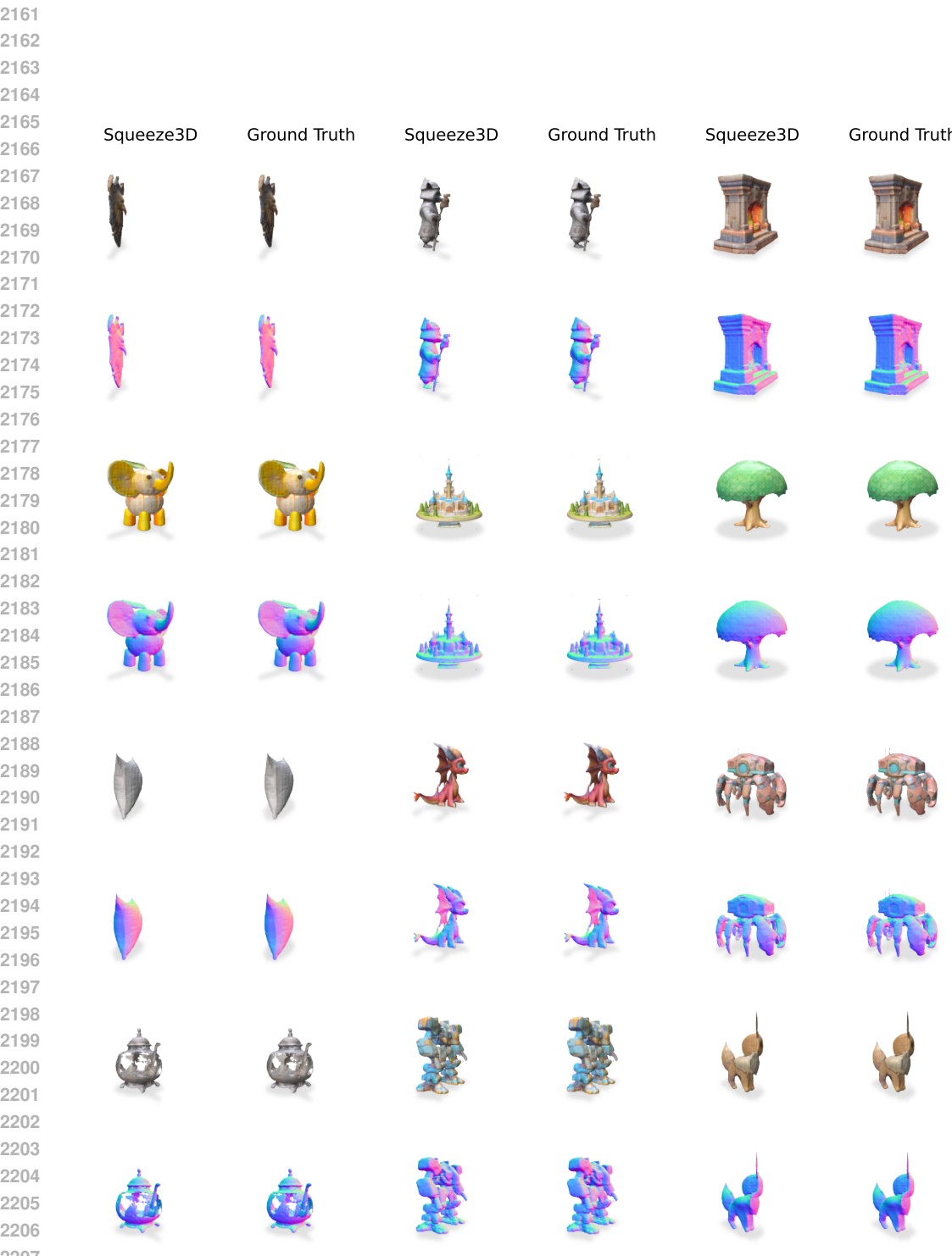

Figure 26: Results Library.

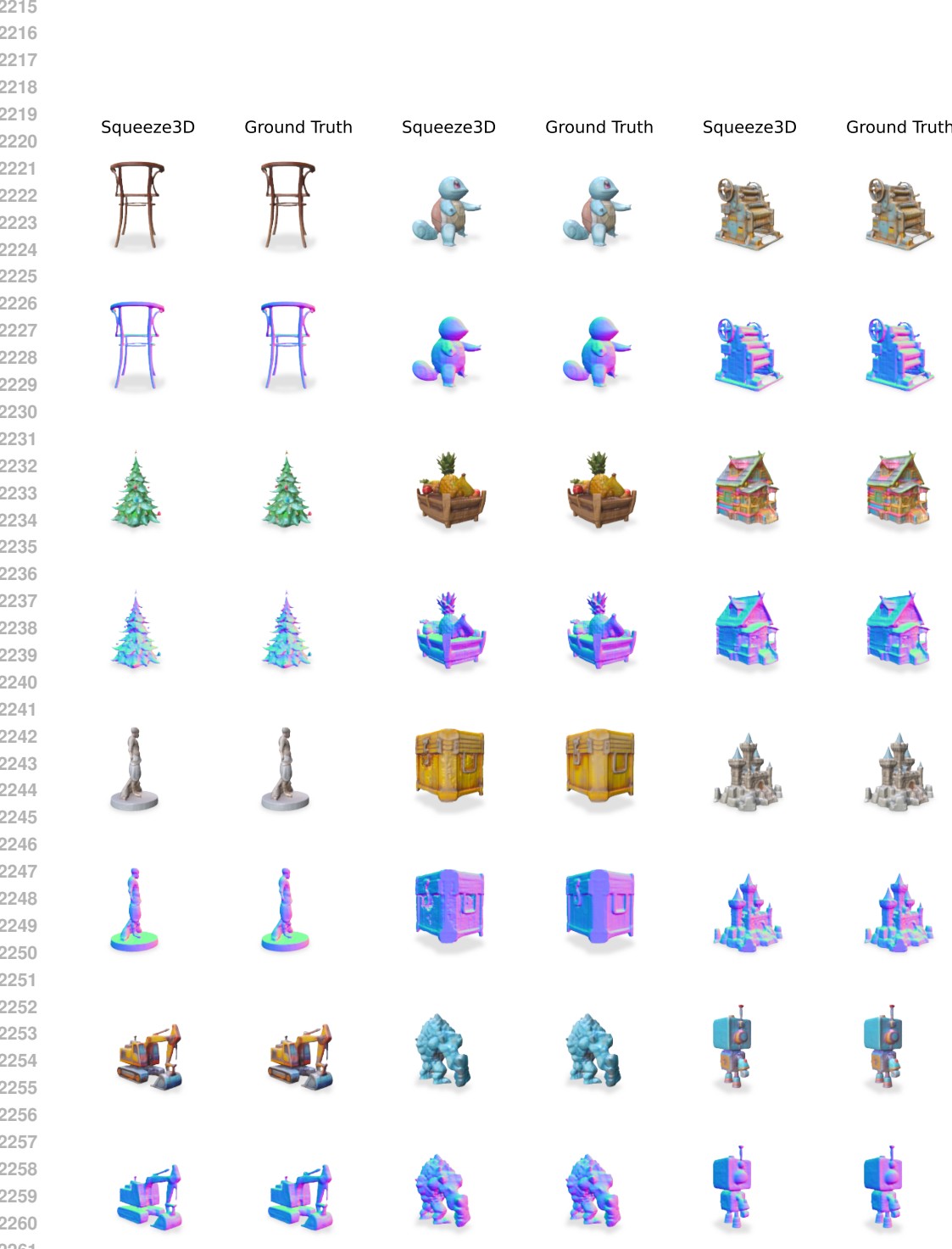

Figure 27: Results Library.

