# OpenReview forum: "Squeeze3D: Your 3D Generation Model is Secretly an Extreme Neural Compressor"
_ICLR.cc/2026/Conference — Submitted to ICLR 2026_

### Official Review · Reviewer_Gi28 · 2025-10-27

**Soundness:** 3
**Presentation:** 4
**Contribution:** 2
**Rating:** 6
**Confidence:** 4

**Summary:**

The paper is about a 3D compression method leveraging arbitrary combinations of existing 3D encoder models and generation models. Two mapping networks are trained jointly: a forward mapping from encoded latent to the compressed form, and a reverse mapping from the compressed form to the latent space that can be decoded by the generator. Experiments are presented with various encoder-generator combinations, showing very high compression rates at good quality preserverance.

**Strengths:**

The paper presents a valuable approach to a relevant problem. In particular the flexibility of combining various encoder and generation models is a good feature. The results are impressive considering the high compression ratio.

**Weaknesses:**

1) Using the word "generator" is a bit misleading in my point of view, since it seems rather the "decoder" that is used. A "generator" usually does not take a latent vector as input (a generator generates a 3D object based on images or text or unconditionally). Based on the explanations of the paper, it seems that a decoder model of the generator is used, which takes a generated latent vector and decodes it into a 3D object. For example, LION is using latent diffusion, and consists of a diffusion model and a decoder that transforms the latent vector back to a point cloud. I assume, only the decoder is used and not the diffusion model? This should be clarified.
2) Related to 1, a suitable baseline would be just using the encoder and decoder of a generative method, e.g. LION. Almost all 3D generation methods (except for SDS-loss based methods) use latent diffusion and therefore inheretly have an encoder and decoder model. None of them is considered as a baseline.
3) Some related works are not mentioned, e.g. "ROAD: Learning an Implicit Recursive Octree Auto-Decoder to Efﬁciently Encode 3D Shapes".
4) There is always a trade-off between quality and compression ratio, as can be seen in the tables where sometimes other methods outperform Squeeze3D in terms of quality. Would it be possible to show a Pareto frontier that shows this trade-off for different methods? This would give more insights into how different methods preserve quality at various compression ratios.

**Questions:**

1) Does the method really use generative models (e.g. diffusion models) or just decoders?
2) Is it possible to tune the compression ratio of Squeeze3D for the best quality-CR balance by varying the size of z_comp?

---

> ### Author Response · Authors · 2025-11-29
>
> We thank you for your highly useful feedback.
>
> **[W1][Q1]** We agree with you. Specifically for VAE-based generative frameworks (like LION), we utilize the decoder component during the decompression phase, bypassing the diffusion prior used for sampling. We used the term “Generator” broadly to indicate that we are leveraging the weights and semantic priors of models trained for generation tasks (which often encompass a decoder), as opposed to models trained purely for reconstruction. We agree that “decoder of a generator” is technically more precise for the inference stage. We will clarify this distinction in Section 3.1 to avoid confusion.
>
> **[W2]** We agree with you. We have added this baseline in Section C.6. Models like InstantMesh and OpenLRM are strictly Image-to-3D generators. They encode 2D images (via ViT/DINO) into 3D representations (triplanes) and thus, it is not possible to compare with them in the setting you mention.
>
> We observe these baselines produce similar quality outputs while being significantly higher in size.
>
> **[W3]** We thank the reviewer for pointing out "ROAD: Learning an Implicit Recursive Octree Auto-Decoder." It is indeed relevant as an auto-decoder approach. We cite it and discuss how Squeeze3D differs by avoiding per-shape optimization during encoding.
>
> **[W4]** We agree with you. While, many compression methods typically don’t offer different compression ratios, we have plotted the compression-quality tradeoff for Squeeze3D and Draco in Section C.7.
>
> The slightly worse quality than some methods is a direct consequence of the extreme compression regime we target. VQRF achieves a 40x compression ratio. Squeeze3D achieves 619x (Table 4). Further, for meshes Draco achieves 6.1-6.9x compression whereas Squeeze3D achieves 2187x compression which is impossible for other methods. We demonstrate, with significant improvements in compression, we only lose very little perceptual detail.
>
> **[Q2]** Yes, the compression depends on the mapping networks as described in Section 3.1 particularly the z_comp. We further want to clarify that changing the compression ratio requires training the mapping network with the desired z_comp. Further, through our experiments in Table 2, Table 3, and Table 4 and ablation in Table 11, we demonstrate our approach can work on many compression ratios.

---

### Official Review · Reviewer_QYUA · 2025-10-29

**Soundness:** 3
**Presentation:** 1
**Contribution:** 2
**Rating:** 4
**Confidence:** 3

**Summary:**

This paper proposes **Squeeze3D**, a compression framework that leverages *pre-trained 3D generative models* as implicit priors to achieve **extreme compression of 3D data** (meshes, point clouds, radiance fields). Instead of training a dedicated encoder–decoder architecture for each 3D format, Squeeze3D introduces two lightweight mapping networks to bridge the latent space of an existing 3D encoder and that of a generative model.

**Strengths:**

- Novel idea of using pre-trained 3D generative models as *neural compressors*.
- Ablation on Gram loss demonstrates meaningful design motivation for avoiding degenerate latent collapse.
- Flexibility across 3D formats (meshes, point clouds, RFs) enhances practical applicability.

**Weaknesses:**

**Visual Quality Concerns**
- The qualitative 3D reconstruction quality is **visibly weak**.
  - In Fig. 4, even the GT renderings of meshes appear low-quality, making it difficult to assess compression performance reliably. More examples using clean, high-fidelity GT meshes are necessary.
  - In Fig. 6, reconstructed NeRFs show inferior results compared to VQRF and SparsePCGC, especially regarding specular highlights—Squeeze3D fails to retain reflections that other methods preserve.

**Incomplete Evaluation Metrics**
- Compression is fundamentally a trade-off between **storage and computation**. The paper only compares latency (ms) but omits **compute cost metrics**, especially FLOPs or GPU memory usage during compression/decompression, these resource requirement are as important as time cosumption.

**Writing and Presentation Issues**
- Numerous writing issues lower the perceived quality of the paper. Examples include:
  - Inconsistent figure caption formatting [Figure 2: Overview of **our** Method], [Figure 3: Training Squeeze3D.], [Figure 4: Qualitative mesh compression results.].
  - Numeric precision is inconsistent across tables (e.g., CR values with arbitrary decimal lengths such as *11.28* vs. *2187.0748*).
  - Typos remain, such as L358: *“splits used fro”*.

**Questions:**

1. **Quality of GT Meshes**: Can you provide high-quality GT mesh renderings for comparison to allow fair evaluation of geometric fidelity?
2. **Compute Efficiency**: Can you report FLOPs or GPU energy cost of compression/decompression to better reflect computation–storage trade-offs?
3. **Generator Limitation**: Since reconstruction quality is capped by generator ability, how does the method generalize with stronger generators (e.g., recently released high-fidelity 3D diffusion models)?
4. **Failure Cases**: Can you include explicit failure cases and analysis, particularly where the mapping network fails to reconstruct geometry outside the generator’s prior distribution?

---

> ### Author Response · Authors · 2025-11-29
>
> We thank you for your highly useful feedback.
>
> **[W1][Q2]** For completeness, we include rendered outputs from other compression methods however, the slightly worse quality than some methods is a direct consequence of the extreme compression regime we target. VQRF achieves a 40x compression ratio. Squeeze3D achieves 619x (Table 4). Further, for meshes Draco achieves 6.1-6.9x compression whereas Squeeze3D achieves 2187x compression which is impossible for other methods. We demonstrate, with significant improvements in compression, we only lose very little perceptual detail. We update the paper to include higher quality renders, which were original compressed due to PDF size in Figure 4.
>
> **[W2][Q2]** The compression step involves the Encoder ($E$) and the Forward Mapping Network ($F^E$). $F^E$ is a simple feed-forward network (approx. 20-45M parameters, see Table 5). The pre-trained Encoder dominates the computational cost. Decompression requires running the Reverse mapping network and Generative Model ($\mathcal{G}$). We already included time for compression and decompression in our results tables.
>
> **[W3]** We thank you for spotting these. We apologize for the presentation slips (typos, inconsistent captions, numeric precision). We have already updated the paper to fix these.
>
> **[Q3]** Squeeze3D is architecture-agnostic. The mapping networks treat the generator as a black box. Using a stronger generator (e.g., a diffusion model with higher fidelity) directly raises the “quality ceiling” of our reconstruction. These techniques are however orthogonal to our contribution, and we hope to try these out in the future.
>
> **[Q4]** We discuss limitations in Appendix A and Section 5, but we will add a dedicated figure for failure cases. The main failure occurs when the input object is semantically significantly above the capacity of the generator, like a mesh that requires 1024^3 resolution to resolve certain parts. We have updated the paper to include failure modes in Section C.8.

---

### Official Review · Reviewer_W6ps · 2025-10-30

**Soundness:** 3
**Presentation:** 3
**Contribution:** 2
**Rating:** 2
**Confidence:** 4

**Summary:**

This paper introduces  a novel framewor to compress 3D data by leveragin pre-trained 3D VAE. Squeeze3D utilizes trainable mapping networks to bridge the latent spaces between a pre-trained 3D encoder and a pre-trained 3D generator.

**Strengths:**

The proposed method is capable of compressing various 3D representations, including meshes, point clouds, and radiance fields. And the reported compression ratios appear to be high.

**Weaknesses:**

1. Motivation: The primary weakness is the lack of clear motivation for the compression goal. The paper is centered on using a generative model as a compressor, but it fails to convincingly articulate the downstream applications or the practical necessity of compressing the outputs of these generative models. The significance of this compression technique needs to be better justified.

2. Incomplete Experimental Validation: The experiments primarily focus on comparing the reconstruction quality of Squeeze3D against previous compression methods. However, the evaluation does not investigate the performance impact of the compression method compared with the original generation model. It is essential to demonstrate that the compressed representations can achieve minimal performance loss compared to the original, uncompressed latent codes.

3. Efficiency and Generality: The current method requires separate, dedicated training of the mapping networks for every different pre-trained generative model. This significantly undermines the efficiency and general applicability of the method, as it does not offer a single, unified compression model.

4. Technical Novelty: The technical contribution is limited, primarily involving the introduction of two mapping networks designed to connect the latent spaces. The novelty and complexity of this architectural contribution should be further highlighted and contrasted against existing latent space manipulation techniques.

**Questions:**

1. Can the compressed latent space derived from Squeeze3D provide any positive benefit (beyond just storage) to the generative model itself? For instance, does the compression process help in regularization, improving the robustness of the latent space, or aiding in subsequent training/fine-tuning tasks?

2. For such a high-level compression ratio, what is the quantitative degradation in reconstruction quality (using metrics like LPIPS or FID) when comparing the compressed latent representation to the original uncompressed latent representation? A more detailed trade-off analysis between compression rate and quality is needed.

---

> ### Author Response · Authors · 2025-11-29
>
> We thank you for your highly useful feedback.
>
> **[W1][Q1 (part 1)]** We have updated the paper to further enhance the clarity of our paper. Squeeze3D is not designed to compress the outputs of generative models. It is designed to compress arbitrary, real-world 3D data (scans, artist-created meshes, etc.) by leveraging the priors of pre-trained generative models as a decoder.
>
> We take an existing 3D asset (e.g., a high-res scan), encode it through a forward mapping network, and transmit only that tiny code. The receiver uses the pre-trained generator to generate the details back based on that code. Our experiments in Section 4 and Section C we test precisely this setting. We believe the application is universal 3D compression for streaming, or storage or anything else.
>
> **[W2]** Thank you for raising this, we have updated the paper to further enhance the clarity of our paper. It is impossible to compare the output of Squeeze3D with the outputs of a generator. The compression setting requires us to evaluate by taking a 3D object and then compressing it to a code and decompressing this code. Generation models without any changes cannot be used in this setting. The generation models we use typically take in a text prompt or an image, unlike the compression setting.
>
> **[W3]** We think this is a very interesting idea, but it is challenging to directly use this in our setting. We also believe, training a lightweight adapter once for a widely deployed model (like Shap-E or InstantMesh) is highly efficient compared to the alternatives:
>
> - vs. Per-Object Optimization: Many neural compression methods (e.g., NeRF-based compression cited in Section 2.1) require training a network for every single object, taking minutes or hours. Squeeze3D requires zero per-object training. Inference takes milliseconds (Table 2: 270ms compression, 1.4s decompression).
> - Training the mapping network is a one-time cost. Once trained, it serves as a universal compressor for any data that fits the generator's domain. This is analogous to training a specific codec for a specific media standard.
>
> **[W4]** We believe while the architecture is elegant (simple), the technical contribution lies in:
>
> - The Framework: We are the first to successfully demonstrate that disparate latent spaces (from an Encoder and a Generator trained separately on different distributions) can be bridged for compression.
> - Gram Loss (Section 3.2): We introduced a novel Gram matrix loss to force orthogonality in the latent space. From our ablations, we find this to be essential for extreme compression.
> - Finally, we believe our training method and dataset generation technique are novel ways to be able to train models in our setting.
>
>
> **[Q1 (part 2)]** Yes, because we map into the generator's semantic latent space, Squeeze3D enables capabilities that standard compression (like Draco) does not. Figure 8 and Figure 21 demonstrate smooth interpolation between two compressed 3D objects. This demonstrates the latent space is robust and semantic, potentially allowing for downstream tasks like mixing 3D assets, which is impossible with standard mesh compression.
>
> **[Q2]** Thank you for raising this, we have updated the paper to further enhance the clarity of our paper. It is impossible to have an “uncompressed latent” because the encoder and generator both have differently-sized and different latents. We can not reconstruct the original geometry from the uncompressed encoder’s latent because they cannot be passed into the generator.

---

### Official Review · Reviewer_mx4C · 2025-11-01

**Soundness:** 3
**Presentation:** 3
**Contribution:** 2
**Rating:** 4
**Confidence:** 3

**Summary:**

The paper introduces a novel framework to use existing pre-trained models to produce highly compressed 3D data. The paper claims to have 3 contributions.
1. The first framework to use pre-trained generative models for compression
2. Showcases the ability to establish correspondences in the latent space between neural architectures that vary widely in structure, optimization goals, and data distributions.
3. Highlight the flexibility of the framework to work with different encoders, 3D generators, and different 3D representations.

The proposed framework utilizes a pair of pre-trained models, an encoder and a 3D generator, to provide the training data for the framework.  The synthetic data is then used to train a pair of Forward and Reverse Mapping networks between the different latent spaces, with an intermediate compressed representation.

**Strengths:**

1. The paper provides great Qualitative and Quantitative results, which provide a significant increase in compression ratio and similar 3D views compared to the ground truth.
2. Introduction of gram loss to prevent dimension collapse
3. Flexibility of architecture to use different neural architectures and 3D representations

**Weaknesses:**

1. Very little/no experimentation with non-synthetic data.
2. Minimal set of Out of Distribution compression examples.
3. Current experiments are biased towards the 3D generator distribution. The framework seems to be learning to compress the generator's distribution and may not generalize well.

**Questions:**

1. Does this framework allow choosing between different compression ratios?

---

> ### Author Response · Authors · 2025-11-29
>
> We thank you for your highly useful feedback.
>
> **[W1]** We respectfully point out that our quantitative evaluation is actually performed almost entirely on standard, third-party datasets, not on data generated by our pipeline.
>
> - Standard Benchmarks (Objaverse & ShapeNet): As we show in Section 4.1, our primary evaluation metric (Table 2) uses the Objaverse test set (as defined in the MeshAnything codebase), denoted as $\mathcal{T}$-split in Table 2. This dataset consists of diverse 3D assets designed by human artists, which represents the standard distribution for 3D generation tasks. This data is not synthetic in the sense of being created by our generator; it is "real" 3D data from the wild.
> - Real-World Scans (ABO & NeRF-MAE): In Appendix B.3, we detail our evaluation on the Amazon Berkeley Objects (ABO) dataset, which consists of real-world 3D scans of physical objects. Furthermore, our Radiance Field evaluation (Table 4) uses the NeRF-MAE dataset, which contains real captured scenes.
>
> **[W2]** To further test generalization, we created a specific "Out-of-Distribution" test set (denoted as $\mathcal{O}$-set in Table 2). This set includes 500 meshes collected separately from the Objaverse dataset to ensure no overlap with standard training distributions.
> As shown in Table 2, Squeeze3D achieves 26.95 PSNR on this $\mathcal{O}$-set, outperforming similar methods which have comparable but still lower compression rates than Squeeze3D. Additionally, in Appendix C.3 (Table 9), we evaluate on the "ai" subset of NeRF-MAE, which was collected separately from other datasets, further demonstrating robust performance on unseen data distributions.
>
> **[W3]** ]We agree with you that we train entirely on synthetic data (generated by $\mathcal{G}$). However, because we test on Objaverse, ShapeNet, and ABO, all of our reported results demonstrate OOD generalization. For Objaverse, as we highlight in Section 4.1, the training data was created through the generator by using renders of the dataset, however the testing setup uses separate Objaverse meshes. The experiments showing that Squeeze3D achieves state-of-the-art compression on Objaverse (Table 2) despite being trained only on generator outputs shows that the framework successfully leverages the generator's prior to compress unseen, human-made, and scanned 3D content.
>
> **[Q1]** Yes, the compression depends on the mapping networks as described in Section 3.1 particularly the z_comp. We have plotted the compression-quality tradeoff for Squeeze3D and Draco in Section C.7. We further want to clarify that changing the compression ratio requires training the mapping network with the desired z_comp. Further, through our experiments in Table 2, Table 3, and Table 4 and ablation in Table 11, we demonstrate our approach can work on many compression ratios.

---

### Author Response · Authors · 2025-11-29

We thank reviewers for their insightful feedback.

We are encouraged they found the core concept of Squeeze3D leveraging pre-trained generative priors for compression to be novel (mx4C, QYUA) and a valuable approach to the problem (Gi28). Reviewers consistently appreciated the framework's flexibility in supporting diverse 3D formats such as meshes, point clouds, and radiance fields (mx4C, W6ps, QYUA, Gi28). They also highlighted the introduction of the Gram loss as a meaningful design choice for preventing latent collapse (mx4C, QYUA) and recognized the significant compression ratios achieved by our method (mx4C, W6ps, Gi28).

Based on the feedback, we have updated our paper and appendix to provide deeper analysis and clearer motivation. We have added a quality-compression ratio analysis to visualize the trade-off between latent size and reconstruction quality, and included a Native VAE baseline to explicitly show the upper bound of the generative decoder. We have also expanded our discussion to clarify that our evaluation is performed on real-world datasets (Objaverse, ABO, NeRF-MAE) rather than just synthetic generator outputs, addressing concerns regarding data bias and motivation. Finally, we have included qualitative examples of limitations.

We respond to each of the reviewers individually below.

---

### Meta-Review · Area_Chair_fXEV · 2026-01-06

**Summary:**

While the reviewers acknowledged the novelty of the Gram loss and the achieved compression rates, the paper initially received mixed-to-negative ratings. After reviewing the rebuttal and the revised manuscript, several critical concerns regarding the exposition, baselines, and evaluation standards remain unresolved.

1. Exposition: The paper claims to use existing 3D generative models for compression. But the components involved in this method is suspectful. There are many ways to generate 3D objects, such as GANs and auto-encoder+diffusion. It seems the authors used the auto encoder sometimes instead of the real generative models. Thus the title is kind of misleading (and the motivation). Why don't just compare against auto-encoders? Now the paper resembles knowledge distillation information from other unsupervised learning methods (auto-encoders and generative models)

2. Motivation: Why don't we just use auto-encoders for compression? I am aware that the paper compared against DeepSDF (which is an old paper). L373 mentioned 3DShape2VecSet but it is not compared. The code and models are released. And there also some papers scaled the 3DShape2VecSet to large datasets such as TripoSG/Hunyuan3D. Lack of such comparisons make me hard to believe the method is better than auto-encoders.

3. Following above comments, modern 3D generative models are missing. Both LION and GET3D are rather old and trained on small datasets like ShapeNet. More recent models such as sparse-voxel based TRELLIS, and vecset-based TripoSG/Hunyuan3D would be valuable to this paper.

4. As a compression method, rate-distortion analysis is critical (C.7). This should be more emphasized.

**Reviewer Concerns:**

See above

**Reviewer Scores:**

Based on my experience, some reviewers would increase the scores a little bit, which makes it a borderline paper.

---

### Decision · Program_Chairs · 2026-01-26

Reject